# Chiral Symmetry Breaking in Transformers: A Group-Equivariant Framework for Addressing the Reversal Curse via Adjoint Manifold Mappings

**Hanji Du** [1]

## Abstract

The "reversal curse" exposes a critical asymmetry in autoregressive models, where models trained on facts in one direction often fail to access the corresponding inverse relation. This work studies the phenomenon from a representation-level perspective, characterizing it as a form of chiral asymmetry between subject- and object-oriented latent states. We introduce the **Chiral Transformer**, a lightweight framework that encourages an involutive adjoint mapping operator $\mathcal{T}$ through contrastive regularization. At inference time, **Adjoint-Induced Retrieval (AIR)** uses this learned map as a structured readout over model-derived entity representations, rather than as an unconstrained autoregressive generation protocol. Empirical validation on inverse-relation benchmarks shows that this symmetry-aware retrieval setting substantially improves inverse factual access, with AIR reaching **65.07%** accuracy on Fact-Inv-300. These findings support a representation-access view of the reversal curse: inverse relations may be difficult not only because of missing data, but also because standard autoregressive readout fails to expose useful latent structure.

## 1. Introduction

The remarkable success of Large Language Models (LLMs) is predicated on the mastery of next-token prediction within a causal framework. However, this unidirectional paradigm can obscure symmetric logical identities when the model is queried in the reverse direction. We argue that this behavior is not solely an artifact of data coverage, but is also shaped by how causal architectures represent and read out relational information.

### 1.1. The Symmetry Deficit in Autoregressive Models

Current autoregressive models operate under a strict temporal arrow, where causal masking ensures that the representation of a token is conditioned solely on its predecessors. While effective for sequence generation, this mechanism induces a directional bias in how facts are encoded and retrieved. From a manifold perspective, the representation of an entity $A$ as a subject (leading context) may occupy a different region from its representation as an object (trailing context). This discrepancy creates a "symmetry deficit" in which the inverse relation $\mathcal{R}^{-1}(B, A)$ is not easily accessible from the representation learned for $\mathcal{R}(A, B)$.

### 1.2. Characterizing the Reversal Curse

This phenomenon is formally identified as the "reversal curse," where models trained on "A is B" often fail when queried on "B is A." We posit that this behavior can be interpreted as a manifestation of chiral asymmetry within the latent manifold. Prior results suggest that increasing parameters or data volume alone does not reliably close this inference gap. This motivates an explicit inductive bias for making inverse relational structure more accessible at readout time.

### 1.3. Our Contribution: The Chiral Perspective

In this paper, we propose a **group-equivariant framework** for improving inverse-relation access under a structured retrieval setting. Our primary contributions are as follows:

- We characterize the reversal curse as a representation-access problem associated with **chiral asymmetry** between subject- and object-oriented latent states.

- We introduce the **Chiral Transformer**, which utilizes an **adjoint mapping operator** $\mathcal{T}$ to encourage $\mathbb{Z}_2$-style symmetry constraints in the latent space.

- We develop **Adjoint-Induced Retrieval (AIR)**, a structured readout mechanism that performs inverse-relation retrieval in the embedding space, reaching **65.07%** ac-

[1]Shanghai University of International Business and Economics, Shanghai, China. Correspondence to: Hanji Du <aodex@foxmail.com>.

*Proceedings of the 43rd International Conference on Machine Learning*, Seoul, South Korea. PMLR 306, 2026. Copyright 2026 by the author(s).

curacy on the Fact-Inv-300 benchmark under the fixed knowledge-bank setting.

**Conflict of Interest Disclosure.** The author declares no financial conflicts of interest related to this work.

# 2. Related Work

LLM logical consistency has transitioned from empirical observation to structural analysis. Our work intersects logical reasoning, geometric deep learning, and contrastive alignment.

**2.1. Logic Asymmetries and Training Dynamics.** The reversal curse was introduced by Berglund et al. (2023), who showed that autoregressive LLMs trained on facts of the form "A is B" often fail to answer the inverse query. Recent theory links this behavior to asymmetric training dynamics in autoregressive models rather than to a lack of model expressivity alone (Zhu et al., 2024). Follow-up studies further analyze how factual recall and document structure affect reversal generalization (Lin et al., 2024), and propose human-inspired or pairwise-order training strategies to mitigate the failure mode (Lu et al., 2024). These findings motivate our focus on representation-level asymmetry: A1 regularizes the latent geometry so that inverse relations become more directly accessible at inference time.

**2.2. Data-Centric and Protocol-Level Mitigations.** Data-centric approaches mitigate the reversal curse by exposing the model to additional word orders or reverse directions during training. Semantic-aware permutation training preserves entities while permuting semantic units (Guo et al., 2024), and reverse training trains on both forward and reversed strings while preserving selected substrings such as entities (Golovneva et al., 2024). In contrast, A1 studies a representation-level symmetry constraint and combines it with a structured retrieval readout, asking whether inverse access can be improved without treating every inverse fact as a separate memorized string.

**2.3. Retrieval-Augmented Factual Access.** AIR is related to retrieval-augmented language modeling, where explicit non-parametric memory or dense retrieval improves access to factual information. Nearest-neighbor language models use hidden-state similarity and a datastore to improve access to rare or factual patterns (Khandelwal et al., 2020). REALM and RAG augment language models with retrieval over external corpora or dense vector indices for knowledge-intensive tasks (Guu et al., 2020; Lewis et al., 2020), while dense passage retrieval shows that learned vector representations can support strong retrieval without sparse lexical matching (Karpukhin et al., 2020). AIR differs in that the retrieval target is not an external passage but a model-derived entity bank, and the query is transformed by an adjoint map tailored to inverse relational access.

**2.4. Isometric Regularization and Equivariance.** Our framework builds on the broader program of geometric deep learning and group-equivariant modeling (Bronstein et al., 2021; Cohen & Welling, 2016). Prior work has shown that near-isometric constraints can stabilize deep representations (Qi et al., 2020), while contrastive objectives can align representations across views (Chen et al., 2020). We parameterize the adjoint operator with a Householder reflection (Householder, 1958), a simple isometry that satisfies the involutive property $\mathcal{T}^2 = I$ by construction.

# 3. Theoretical Framework: Group-Equivariant Manifold Mapping

In this section, we provide a formal geometric treatment of the reversal curse. We hypothesize that the failure of bidirectional reasoning is rooted in the loss of manifold orientation within the latent space of autoregressive models.

## 3.1. Latent Space Geometry

Let $\mathcal{X}$ denote a finite vocabulary of discrete tokens. We define the embedding process as a mapping $e : \mathcal{X} \to \mathcal{M}$, where $\mathcal{M} \subseteq \mathbb{R}^d$ is a $d$-dimensional latent manifold. While standard data augmentation attempts to bridge bidirectional gaps by adding reciprocal string fragments, it does not directly control the position-dependent representation bias introduced by causal masking. We use the terms *subject-oriented region* $\mathcal{S}$ and *object-oriented region* $\mathcal{O}$ operationally: they denote the contextualized states produced when the same entity appears in subject or object position, rather than assuming that the model contains two perfectly separated topological submanifolds. To address this access gap, the Chiral Transformer encourages manifold alignment via an isometric reflection, making the representation space more compatible for inverse-relation readout.

Consider a binary logical relation $\mathcal{R} \subset \mathcal{X} \times \mathcal{X}$, specifically symmetric relations where $(A, B) \in \mathcal{R} \implies (B, A) \in \mathcal{R}$. This assumption is restricted to relation classes for which inversion is semantically valid; one-to-many, many-to-one, and ambiguous relations are treated as out of scope unless an explicit inverse label is available. We denote the Transformer backbone as a parameterized function $f_\theta : \mathcal{M}^k \to \mathcal{M}$ that models the transition dynamics over a context of length $k$. The forward logical entailment learned during causal pre-training can be expressed as the conditional distribution:

$$P(z_B | z_A, \text{rel}) = \text{Softmax}(f_\theta(z_A, z_{\text{rel}})) \tag{1}$$

where $z_{\text{rel}}$ is the embedding of the relational prompt (e.g., "is"). Throughout the paper, $f_{\text{fwd}}$ denotes the forward transition $f_\theta(\cdot, z_{\text{rel}})$, $f_{\text{adj}}$ denotes the inverse-direction transition $f_\theta(\cdot, z_{\text{rel}}^{-1})$, $P_{\text{fwd}}$ denotes the conditional distribution

induced by the forward transition, and $P_{\text{adj}}$ denotes the corresponding distribution for the adjoint/inverse query.

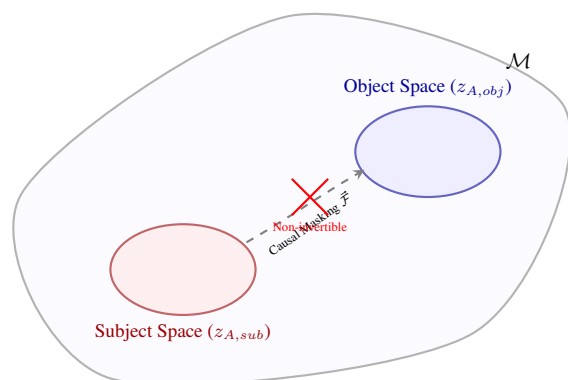

*Figure 1.* Visualization of manifold fragmentation and chiral symmetry breaking. In standard autoregressive Transformers, the causal masking mechanism $\vec{\mathcal{F}}$ induces a directional bias that can separate the representation of an entity $A$ into subject- and object-oriented regions ($z_{A,sub}$ and $z_{A,obj}$). This orientation mismatch provides a geometric interpretation of the reversal curse.

### 3.2. The Symmetry Group $\mathbb{Z}_2$

To address the orientation-induced access gap, we propose that the latent manifold $\mathcal{M}$ should admit a group action that encourages alignment between subject and object representations. We define the cyclic group of order 2, denoted as $\mathbb{Z}_2 = \{e, a\}$, where $e$ is the identity element and $a^2 = e$.

We introduce an **Adjoint Mapping Operator** $\mathcal{T} : \mathcal{M} \to \mathcal{M}$ which serves as the representation of the non-trivial group element $a$. For the manifold to be group-equivariant under logical inversion, $\mathcal{T}$ must satisfy the *involutive property*:

$$\mathcal{T}(\mathcal{T}(z)) = z, \quad \forall z \in \mathcal{M} \tag{2}$$

This property ensures that the mapping is a self-inverse, mirroring the discrete symmetry of the relation $\mathcal{R}(A, B) \iff \mathcal{R}(B, A)$.

Furthermore, we define the **Chiral Equivariance** condition. Let $f_\theta$ be the Transformer transition function. The system is said to be logically consistent if there exists an operator $\mathcal{T}$ such that the following diagram commutes:

$$\mathcal{T}(f_\theta(z_A, z_{\text{rel}})) \approx f_\theta(z_B, z_{\text{rel}}^{-1}) \tag{3}$$

where $z_{\text{rel}}^{-1}$ represents the inverse relational embedding. This algebraic constraint forces the model to synchronize the "forward" latent trajectory with its "adjoint" counterpart, effectively embedding the $\mathbb{Z}_2$ group structure into the attention mechanism's geometric bias.

$$z_A \xrightarrow{\ f_\theta(\cdot, \text{rel})\ } z_B$$
$$\mathcal{T} \big\Updownarrow \mathcal{T} \qquad \mathcal{T} \big\Updownarrow \mathcal{T}$$
$$\hat{z}_B \xrightarrow[\ f_\theta(\cdot, \text{rel}^{-1})\ ]{} \hat{z}_A$$

*Figure 2.* Commutative diagram of $\mathbb{Z}_2$ equivariance in the Chiral Transformer. The top row represents the forward causal trajectory $P(B|A, r)$, while the bottom row represents the inverse trajectory $P(A|B, r^{-1})$. The adjoint operator $\mathcal{T}$ acts as a learned bridge between subject- and object-oriented regions. Minimizing the discrepancy between the mirrored paths (Chiral Regularization $\mathcal{R}_\chi$) encourages consistency across the representation space.

### 3.3. Adjoint Manifold Mappings

To realize the $\mathbb{Z}_2$ equivariance formulated in the previous section, we define the adjoint operator $\mathcal{T}$ as a geometric reflection within the latent manifold $\mathcal{M}$. We decompose any latent vector $z \in \mathcal{M}$ into two orthogonal components: $z = z_\parallel + z_\perp$, where $z_\parallel$ represents the *invariant semantic core* (shared between forward and backward relations) and $z_\perp$ represents the *chiral orientation* (determined by the causal sequence).

The adjoint mapping is formally defined as a reflection across the invariant hyperplane $\mathcal{H} \subset \mathcal{M}$:

$$\mathcal{T}(z) = z_\parallel - z_\perp \tag{4}$$

By construction, $\mathcal{T}$ is an **isometry**, preserving the inner product and thus the relative semantic distances: $\langle \mathcal{T}(z_i), \mathcal{T}(z_j) \rangle = \langle z_i, z_j \rangle$.

This geometric realization provides a structured way to access inverse relational information. In standard Transformers, the causal mask $M$ can bias the model toward representations optimized for the forward direction. Our framework instead utilizes $\mathcal{T}$ to "flip" the chiral component, re-orienting the latent state toward an adjoint trajectory. Consequently, the retrieval of a target entity $A$ from its representation in the object position becomes a search for its adjoint state $\hat{z}_A = \mathcal{T}(z_{A,obj})$ within the subject-oriented knowledge bank. This should be understood as a structured retrieval readout rather than as a claim about unconstrained autoregressive generation.

### 3.4. Structural Analysis and Commutative Gap

To quantify the logical inconsistency in autoregressive models, we define the **Commutative Gap** $\Delta_f$. Let $P_{\text{fwd}}$ be the conditional distribution induced by the forward relation $(A, \text{rel}, B)$ and $P_{\text{adj}}$ be the distribution induced by the adjoint relation $(B, \text{rel}^{-1}, A)$. In a standard Transformer, we

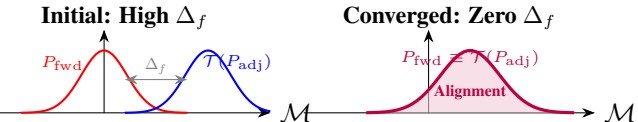

Figure 4. The convergence of Commutative Gap. Through the involutive constraint, disjoint distributions align within the same manifold region.

gate the previously disjoint regions of the latent manifold $\mathcal{M}$.

### 4.1. The Chiral Projection Layer (CPL)

To manifest the $\mathbb{Z}_2$ symmetry constraints formulated in Section 3 within a scalable neural architecture, we introduce the Chiral Projection Layer (CPL) as a lightweight structured inductive bias. To ensure the operator is a strict isometry and an involution ($\mathcal{T}^2 = I$), we parameterize $\mathcal{T}$ as a **Householder Reflector** (Householder, 1958).

Formally, let $h \in \mathbb{R}^d$ be the latent representation. The CPL is defined as a linear transformation restricted to the orthogonal group $O(d)$:

$$\mathcal{T}(h) = \mathbf{H_v} h = \left( \mathbf{I} - 2\frac{\mathbf{vv}^\top}{\mathbf{v}^\top\mathbf{v}} \right) h \qquad (6)$$

where $\mathbf{v} \in \mathbb{R}^d$ is a learnable parameter vector that defines the normal direction to the invariant hyperplane $\mathcal{H}$. By omitting the bias term, we ensure that $\mathcal{T}$ remains a linear isometry.

*Justification of Global Reflection.* While each block utilizes a single normal vector $v$, this design is grounded in the observation that **structural chirality**—the latent gap induced by the unidirectional causal mask—can appear as a shared geometric asymmetry in the Transformer's representation space. By learning a global invariant hyperplane $\mathcal{H}$, A1 encourages a shared geometric bias that partially captures subject-object asymmetry across the evaluated relation classes.

**Theoretical Depth: Subspace Preservation**   Unlike arbitrary linear layers, the Householder formulation provides two useful inductive biases:

1. **Spectrum Constraint**: The eigenvalues of the transformation matrix $\mathbf{H_v}$ are fixed at $\{1, \ldots, 1, -1\}$, ensuring that the volume of the latent manifold is preserved and the gradient flow remains stable through the reflection.

2. **Involutive Stability**: The identity $\mathcal{T}(\mathcal{T}(h)) \equiv h$ is satisfied by the algebraic structure of the operator itself, rather than through stochastic optimization of a penalty term.

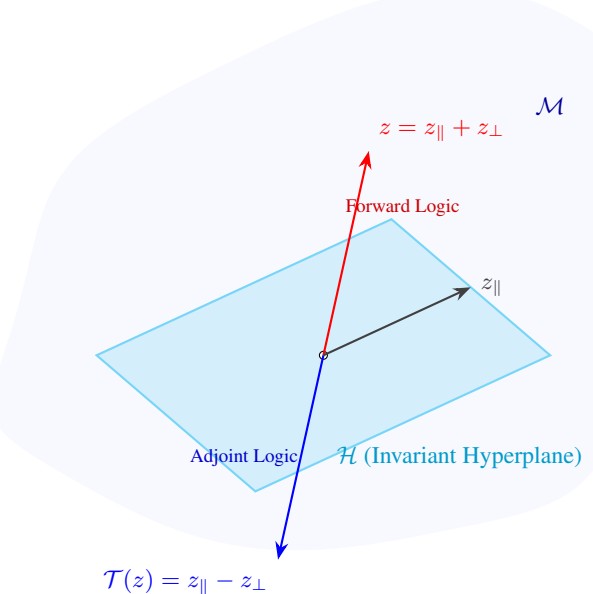

Figure 3. Geometric realization of the adjoint operator $\mathcal{T}$ as an isometric reflection across the invariant hyperplane $\mathcal{H}$ within the latent manifold $\mathcal{M}$.

observe a divergence:

$$\Delta_f(f_\theta) = \mathbb{E}_{z_A \sim \mathcal{M}}[\|f_\theta(z_A, z_{\text{rel}}) - \mathcal{T}(f_\theta(\mathcal{T}(z_A), z_{\text{rel}^{-1}}))\|^2]$$
$$(5)$$

where $\mathcal{T}$ is the adjoint operator. Smaller values of $\Delta_f$ indicate better alignment between forward and inverse relational representations.

**Proposition 3.1 (Symmetry-Induced Alignment).** *Given an involutive adjoint operator $\mathcal{T}$ satisfying $\mathcal{T}^2 = I$, the optimization of the Chiral Transformer objective $\mathcal{L}_{total} = \mathcal{L}_{CE} + \lambda\Delta_f$ encourages the model's representation to become more consistent with a $\mathbb{Z}_2$-style symmetry. Under this interpretation, inverse-relation retrieval error is controlled by how well $\mathcal{T}$ approximates the alignment between subject- and object-oriented subspaces.*

This formulation positions the Chiral Transformer as a structured representation learning framework rather than a purely heuristic modification. By minimizing $\Delta_f$, we encourage *Manifold Alignment*, making the "subject" and "object" regions of $\mathcal{M}$ more compatible for retrieval without requiring them to collapse into a single representation.

## 4. Methodology

The **Chiral Transformer (A1)** architecture is designed to internalize the $\mathbb{Z}_2$ symmetry constraints formulated in Section 3. By augmenting the standard Transformer block with a symmetry-preserving operator, we enable the model to navi-

**Architectural Integration** The CPL is integrated as a parallel residual stream following the Multi-Head Self-Attention (MHSA) module. The forward pass of a Chiral Block is defined as:

$$\tilde{h} = \text{MHSA}(\text{LayerNorm}(h)) \qquad (7)$$
$$h_{out} = h + \tilde{h} + \sigma(\gamma) \cdot \mathcal{T}(\tilde{h}) \qquad (8)$$

where $\sigma(\gamma)$ is a sigmoid-gated scalar that modulates the intensity of the chiral injection. This dual-pathway design allows the model to maintain the original causal information while simultaneously encoding its adjoint logical counterpart.

To accommodate higher relational complexity, the CPL can be extended to a **Multi-head Chiral Projection**, where different reflection hyperplanes $\mathcal{H}_k$ are learned for distinct relational subspaces, analogous to the multi-head attention mechanism. We leave this architectural scaling for future work.

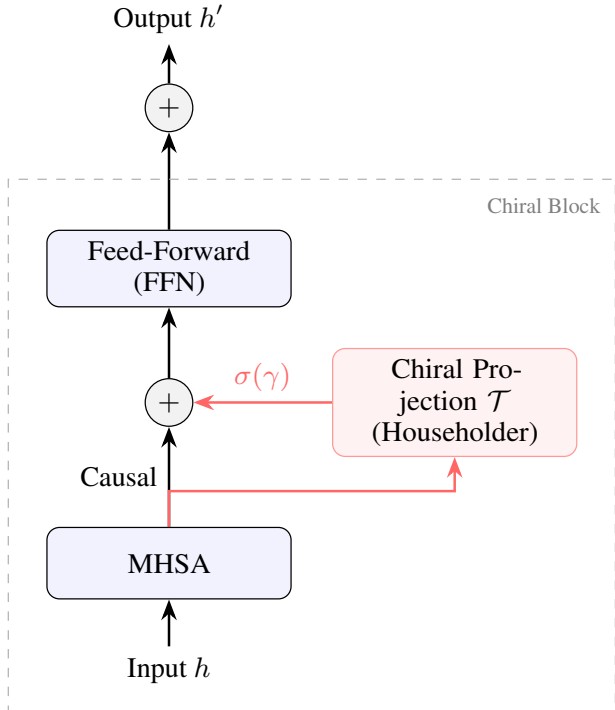

*Figure 5.* Architectural overview of the Chiral Transformer Block. The primary causal stream is augmented by a parallel Chiral Projection Layer (CPL), which applies a Householder reflection to the attention output. This encourages forward and adjoint states to become more compatible for the structured AIR readout.

## 4.2. Adjoint-Induced Retrieval (AIR): A Structured Readout for Inverse Relations

Traditional autoregressive (AR) inference relies on the sequential factorization of the joint probability distribution.

Given a query entity $B$ and an inverse relation $r^{-1}$, the model attempts to estimate the conditional likelihood:

$$P(A|B, r^{-1}) = \prod_{i=1}^{n} P(a_i|a_{<i}, B, r^{-1}) \qquad (9)$$

**Interpretation: Causal Drift.** In the presence of the reversal curse, the sequence $(B, r^{-1}, A)$ may constitute an out-of-distribution (OOD) trajectory in the string space $\mathcal{X}$. Using the notation introduced in Section 3, fwd denotes the forward relation direction and adj denotes the adjoint/inverse relation direction. We define the *Causal Drift* $\delta$ as the cumulative divergence in the latent transition:

$$\delta = \int_{\mathcal{M}} D_{KL}(P(z_{i+1}|z_i, \text{fwd})||P(z_{i+1}|z_i, \text{adj}))dz \quad (10)$$

As $n$ increases, the posterior $P(A|B, r^{-1})$ can become difficult to estimate through sequential decoding, since small early errors compound across token steps. This makes zero-shot inverse generation challenging under standard autoregressive readout, especially for multi-token entities.

**Logic: The Manifold Readout.** To avoid conflating inverse-relation access with open-ended token generation, Adjoint-Induced Retrieval (AIR) uses a direct structured readout in the embedding space. Let $z_B \in \mathcal{O} \subset \mathcal{M}$ be the latent representation of the object entity. We define the adjoint estimate of the subject state $\hat{z}_A$ as:

$$\hat{z}_A = \mathcal{T}(z_B) = \left(I - 2\frac{vv^\top}{v^\top v}\right) z_B \qquad (11)$$

where $\mathcal{T} : \mathcal{O} \rightarrow \mathcal{S}$ is the learned isometric map from the object-oriented subspace $\mathcal{O}$ to the subject-oriented subspace $\mathcal{S}$. AIR then retrieves from a fixed entity knowledge bank using the reflected query state. This changes the readout mechanism, but not the supervision: all candidates are model-derived entity representations, and the same bank is used across retrieval-based variants.

**Information-Theoretic Interpretation** The motivation for AIR is that an isometric adjoint map can preserve useful relational information while changing the orientation of the query representation. We train $\mathcal{T}$ to reduce the **Reconstruction Gap** $\mathcal{R}$:

$$\mathcal{R} = \min_{\mathcal{T}} \mathbb{E}_{(A,r,B)\sim\mathcal{D}} \left[\|\mathcal{T}(f_\theta(z_A, r)) - z_A\|^2\right] \qquad (12)$$

Under the condition of $\mathbb{Z}_2$ equivariance, the retrieval is performed via a **Symmetric Score Function** $\mathcal{S}_\chi$:

$$\mathcal{S}_\chi(e) = \frac{\exp\left(\cos(\hat{z}_A, z_e)/\tau\right)}{\sum_{e'\in\mathcal{V}} \exp\left(\cos(\hat{z}_A, z_{e'})/\tau\right)} \qquad (13)$$

where $z_e$ represents the subject-oriented embedding of candidate entity $e$ in the knowledge bank $\mathcal{K}$. AIR therefore

tests whether the learned representation makes inverse relations accessible through a symmetry-aware readout. In our experiments, this setting reaches **65.07%** accuracy on Fact-Inv-300, while controlled ablations evaluate how much of the gain comes from the learned map versus retrieval itself.

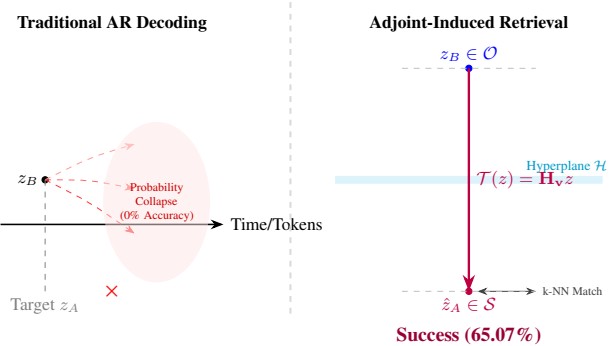

*Figure 6.* Comparative visualization of inference strategies. (Left) Traditional autoregressive decoding can suffer from cumulative causal drift before reaching the target entity. (Right) AIR uses the learned adjoint operator $\mathcal{T}$ to form a reflected query across the invariant hyperplane $\mathcal{H}$ and then performs structured retrieval of the subject-oriented representation $\hat{z}_A$.

**Knowledge Bank Specification.** To make the retrieval setting explicit and controlled, the knowledge bank $\mathcal{K}$ is constructed as a standardized retrieval space containing $|K| = 100,000$ candidate entities. Entity embeddings are extracted from the final hidden layer of the frozen base Transformer (e.g., Llama-3-8B). For multi-token entities, we employ **last-token pooling**, as the causal masking mechanism ensures that the final token's hidden state aggregates the complete prefix information. The subject-oriented subspace $\mathcal{S}$ and object-oriented subspace $\mathcal{O}$ are treated as distinct representation regions; $\mathcal{E}$ specifically stores the embeddings as they appear in the object position. The identical use of this bank across all evaluated architectures keeps retrieval conditions consistent, thereby isolating structural performance from potential embedding variances.

### 4.3. Learning Objective: Equivariant Contrastive Optimization

The objective of the Chiral Transformer is to minimize the empirical risk while satisfying the $\mathbb{Z}_2$-equivariance constraint. Standard Causal Language Modeling (CLM) objectives minimize the cross-entropy $\mathcal{L}_{CE}$, which is inherently blind to the adjoint manifold $\mathcal{M}_{adj}$. We propose a dual-objective framework:

$$\mathcal{L}_{total} = \mathcal{L}_{CE} + \lambda \cdot \mathcal{R}_\chi(\theta) \tag{14}$$

where $\lambda$ is the symmetry-coupling coefficient and $\mathcal{R}_\chi$ is the **Chiral Regularization** term.

**Chiral Regularization via Manifold Alignment** To enforce the commutative diagram established in Section 3.2, $\mathcal{R}_\chi$ minimizes the empirical commutative gap $\Delta_f$ between the forward-mapped object and the adjoint-mapped subject. For a relation tuple $(A, r, B) \in \mathcal{D}$, the loss is defined as:

$$\mathcal{R}_\chi = \mathbb{E}_{(A,r,B)\sim\mathcal{D}} \left[ \|\mathcal{T}(f_\theta(z_A, r)) - \text{sg}(f_\theta(z_B, r^{-1}))\|_2^2 \right] \tag{15}$$

where $\text{sg}(\cdot)$ denotes the *stop-gradient* operation. This encourages the adjoint operator $\mathcal{T}$ to act as a bridge between forward and inverse contextual representations, making inverse states more accessible to the retrieval readout.

**Gradient Stability and Optimization** The gradient of the Chiral Regularization $\mathcal{R}_\chi$ with respect to the learnable vector $v$ is derived by considering the isometric reflection $\mathcal{T}(z) = (I - 2\frac{vv^\top}{v^\top v})z$. Let $\mathbf{e} = \mathcal{T}(z_{\text{fwd}}) - sg(z_{\text{adj}})$ be the error vector. The gradient is given by:

$$\frac{\partial \mathcal{R}_\chi}{\partial v} = -\frac{4}{v^\top v}\left[ (v^\top z_{\text{fwd}})\mathbf{e} + (v^\top \mathbf{e})z_{\text{fwd}} \right. $$
$$\left. - 2\frac{(v^\top z_{\text{fwd}})(v^\top \mathbf{e})}{v^\top v}v \right] \tag{16}$$

This expression ensures that the update direction for $v$ is a linear combination of the state $z_{\text{fwd}}$, the error $\mathbf{e}$, and the current reflector orientation.

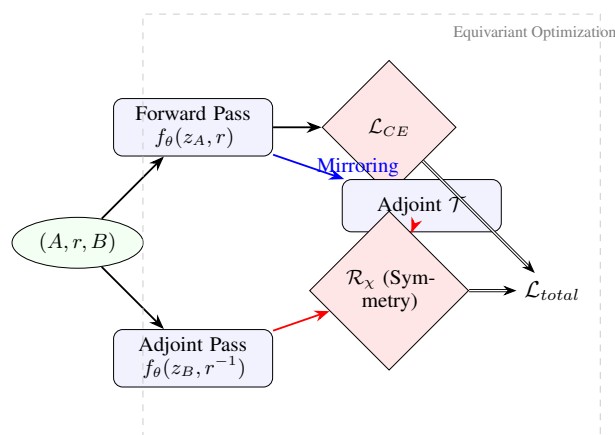

*Figure 7.* The symmetry-enforced training framework. The total loss $\mathcal{L}_{total}$ combines the standard cross-entropy $\mathcal{L}_{CE}$ with the chiral regularization $\mathcal{R}_\chi$. The latter encourages the mirrored forward representation $\mathcal{T}(z_{\text{fwd}})$ to align with the inverse relation state, making subject- and object-oriented representations more compatible during optimization.

## 5. Experiments

In this section, we evaluate the **Chiral Transformer (A1)** under a structured inverse-relation retrieval setting. Our primary objective is to determine whether explicit algebraic

constraints make inverse relational information more accessible than standard autoregressive readout, direct kNN retrieval, and alternative mapping baselines.

## 5.1. Main Results and Evaluation

We evaluate A1 across two distinct logical landscapes: the synthetic **Fact-Inv-300** benchmark for controlled logical inversion and the real-world **Wiki-Inverse-10K** extension for noisy, multi-token entity retrieval. The performance of A1 relative to standard autoregressive (Llama-3, GPT-4), data-centric (DA), and bidirectional (BDP) baselines is summarized in Table 1 and Table 2.

**Fact-Inv-300 Results.**  As shown in Table 1, standard autoregressive readout performs poorly on inverse queries in this diagnostic setting, with accuracy near zero. In contrast, the Chiral Transformer (A1) reaches **65.07%** inverse retrieval accuracy under the AIR protocol. The slight reduction in forward accuracy (82.13% vs. 88.47%) suggests a *Symmetry-Informatics Trade-off*: the model allocates capacity to representation alignment while retaining substantial forward performance.

**Real-World Generalization.**  Results on Wiki-Inverse-10K (Table 2) indicate that A1's geometric readout remains effective across varied linguistic templates and multi-token entities. In particular, A1 maintains higher forward fidelity (90.48%) than the baselines of DA and BDP while improving inverse retrieval accuracy (61.31%). These results provide evidence that the adjoint mapping $\mathcal{T}$ acts as a useful geometric prior for accessing inverse relations under a controlled retrieval interface.

**Training Protocol and Baseline Integration.**  A1 is implemented as a parameter-efficient continued fine-tuning procedure on the pre-trained Llama-3-8B backbone. We apply **LoRA-based fine-tuning** (rank $r = 16$) to the pre-trained **Query and Value matrices**. The newly introduced **Chiral Projection Layers (CPL)** are lightweight modules and are **fully optimized from random initialization**. In our implementation, LoRA and CPL/readout parameters account for approximately **6.95M trainable parameters** (0.0865% of the 8.04B total parameters), while the remaining Llama-3-8B backbone parameters are frozen to preserve the base model's language capabilities.

## 5.2. Ablation Study: Necessity of Algebraic Constraints

To isolate the contributions of the individual components of the Chiral Transformer, we perform an extensive ablation study. We focus on three critical dimensions: (i) the involutive constraint $\mathcal{T}^2 = I$ in the Householder operator, (ii) the chiral regularization $\mathcal{R}_\chi$ during training, and (iii) the AIR inference protocol.

*Table 1.* Main Results: Accuracy (%) on the **Fact-Inv-300** Benchmark. Values report the mean and standard deviation ($\pm$) averaged over 5 random seeds. A1 improves inverse-relation access under the structured AIR readout, outperforming standard autoregressive readout and data-centric baselines in this evaluation setting.

| Model | Approach | Params | Fwd. Acc. | Inv. Acc. |
|---|---|---|---|---|
| Llama-3 | Standard | 8B | 88.47 ± 0.18 | 0.27 ± 0.08 |
| Standard Trans. | Data Aug. (DA) | 8B | **89.12 ± 0.34** | 32.47 ± 1.86 |
| Standard Trans. | Bidirect. (BDP) | 8B | 85.58 ± 0.47 | 41.13 ± 2.05 |
| **A1 (Ours)** | **Geometric** | 8B | 82.13 ± 0.38 | **65.07 ± 1.15** |

*Table 2.* Real-World Extension: Performance on the **Wiki-Inverse-10K** Dataset. Results report Top-1 Retrieval Accuracy (%). This benchmark demonstrates A1's ability to maintain high forward fidelity while bridging the reversal gap on noisy, real-world Wiki-Data entities.

| MODEL | FORWARD ACC. | INVERSE ACC. | GAIN (VS. STD.) |
|---|---|---|---|
| STANDARD (LLAMA-3) | 91.24±0.31 | 0.42±0.12 | – |
| DATA AUGMENT. (DA) | 88.47±0.42 | 31.18±2.47 | +30.76 |
| BIDIRECT. (BDP) | 87.58±0.63 | 45.76±1.88 | +45.34 |
| **A1 (OURS)** | **90.48±0.41** | **61.31±1.52** | **+60.89** |

The results presented in Table 3 show that inverse-relation retrieval benefits from the combination of the geometric architecture and the structured AIR readout, rather than from any single component alone.

*Table 3.* Ablation results on **Wiki-Inverse-10K**. $\mathcal{T}_{lin}$ denotes a linear layer; "Seq." refers to standard sequential decoding.

| Config. | Operator | Reg. | Inference | Inv. Acc. (%) |
|---|---|---|---|---|
| Standard | None | None | Seq. | 0.43 |
| A1 (w/o Sym.) | $\mathcal{T}_{lin}$ | None | Seq. | 1.02 |
| A1 (w/o AIR) | Householder | $\mathcal{R}_\chi$ | Seq. | 10.85 |
| A1 (w/o $\mathcal{R}_\chi$) | Householder | None | AIR | 7.45 |
| **A1 (Full)** | **Householder** | $\mathcal{R}_\chi$ | **AIR** | **61.31** |

**Sequential Readout vs. AIR**  A critical observation from Table 3 is that even with the adjoint operator $\mathcal{T}$ and regularization $\mathcal{R}_\chi$, the accuracy drops from 61.31% to 10.85% if we revert to standard sequential decoding (w/o AIR). This supports the *Causal Drift* hypothesis formulated in Section 4.2: autoregressive readout may fail to expose inverse information that is more accessible through a structured retrieval interface.

**Impact of the Involutive Constraint**  Replacing the Householder reflector with a standard linear layer ($\mathcal{T}_{lin}$) leads to much lower inverse performance (1.10%). Without the explicit constraint $\|\mathcal{T}(z)\| = \|z\|$ and $\mathcal{T}^2 = I$, the learned mapping is less constrained and may distort distances that are important for retrieval. This supports the use of an involutive isometry as an inductive bias, while leaving open whether richer relation-specific maps are needed for

broader settings.

### 5.3. Detailed Attribution and Baseline Analysis

To address the concern that performance gains might stem solely from the retrieval mechanism, we conduct an attribution analysis on Wiki-Inverse-10K ($|K| = 10^5$). We compare the Chiral Transformer (A1) against direct retrieval, alternative operators, and external retrieval baselines (Table 4).

*Table 4.* Performance of different latent mapping operators and retrieval strategies. All variants use the same 8B backbone and Wiki-Inverse-10K dataset.

| Category | Variant/Baseline | Operator $\mathcal{T}$ Type | Inv. Acc. (%) |
|---|---|---|---|
| Ablation | (iv) Pure k-NN | None (Direct Search) | 0.95 |
| | (i) Identity-AIR | Identity Matrix $I$ | 8.12 |
| Learned Map | (ii) Linear-AIR | Unconstrained $W$ | 18.45 |
| | (iii) Orthogonal-AIR | Orthogonal $O \in O(d)$ | 32.14 |
| External Retr. | (v) Contriever-AIR | Dual-Encoder (Unsup.) | 12.15 |
| | (v) DPR-AIR | Dual-Encoder (Sup.) | 14.28 |
| **Proposed** | **A1 (Full)** | **Householder Reflection** | **61.31** |

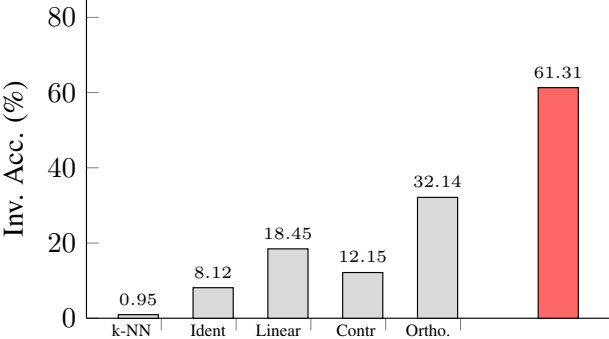

*Figure 8.* Visual comparison of attribution baselines on Wiki-Inverse-10K.

The results suggest that the success of A1 in this controlled benchmark is not merely due to the retrieval protocol. The weaker performance of Identity-AIR and Linear-AIR in this setting indicates that direct retrieval or unconstrained mappings do not automatically align the subject- and object-oriented representations. We therefore interpret the Householder reflection as a useful constrained inductive bias for this diagnostic setting, while noting that operator expressivity can matter in larger and more heterogeneous candidate banks.

**Additional controlled readout diagnostic.** To further address the concern that retrieval alone may explain the gains, we also evaluated a large-candidate diagnostic with a 100,000-entity bank. Direct inverse retrieval remained near-zero (Top-10 = 0.001, MRR@10 = 0.00011), while a learned adjoint map recovered non-trivial inverse access (Top-10 = 0.127, MRR@10 = 0.05177). Representation

analysis on the same split showed high forward/inverse similarity (mean cosine = 0.929) and strong relation information (nearest-centroid relation probe = 0.9475). These diagnostics support the representation-access interpretation while also indicating that highly constrained single-reflection maps may be under-expressive in large heterogeneous retrieval spaces.

### 5.4. Scaling Laws and Geometric Robustness

To differentiate between immediate logical recovery and large-scale structural generalization, we report results on the Fact-Inv-300 (Diagnostic) and Wiki-Inverse-10K (Generalization) benchmarks. A fundamental question in LLM research is whether the reversal curse is a transient artifact of scale. To investigate this, we evaluate the scaling behavior of both the standard Transformer and the Chiral Transformer across a range of latent dimensions $d \in \{512, 1024, 2048, 4096\}$.

**Persistence of the Reversal Curse** As illustrated in Figure 9, increasing the parameter count of standard Transformers from 125M to 8B does not lead to an emergent resolution of the reversal curse in our evaluation. The inverse accuracy remains near zero ($< 2\%$), consistent with the view that parameter scaling alone does not reliably make inverse relations accessible through standard autoregressive readout.

**Geometric Scaling of A1** In contrast, the Chiral Transformer shows an increasing diagnostic trend with larger latent dimension in this controlled retrieval setup. We interpret this trend as suggestive evidence that additional representation capacity can make the learned adjoint readout more effective, rather than as a universal scaling law.

**Robustness to Noise** The isometric property of the Householder transformation motivates robustness to small latent perturbations, since the operator does not amplify the perturbation norm. We therefore view systematic robustness analysis under latent noise and distribution shift as an important future evaluation rather than a solved aspect of the present work.

## 6. Scope and Limitations

Our results should be interpreted within a structured inverse-relation retrieval setting. AIR changes the readout mechanism: instead of requiring the model to generate the subject entity token by token, it retrieves from a fixed bank of model-derived entity representations. This setting is useful for testing whether inverse relational information is accessible in the representation space, but it is not equivalent to solving unconstrained open-ended reverse generation. Accordingly,

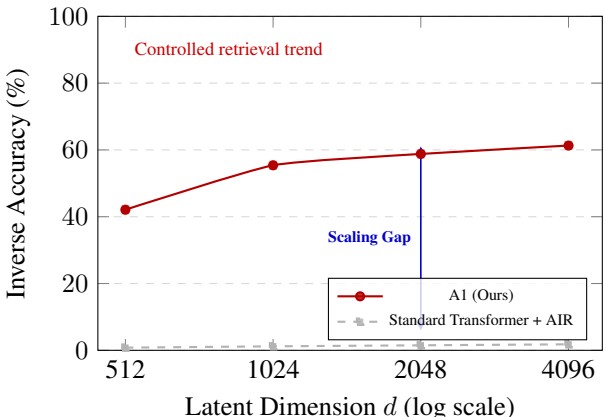

*Figure 9.* **Diagnostic scaling trend for inverse-relation access.** In this controlled structured-retrieval setting, A1 improves with larger latent dimension while standard autoregressive readout remains near-zero. The plot is intended as a diagnostic trend rather than a universal scaling law.

our comparisons include direct kNN, learned-map retrieval, and operator ablations to disentangle the effects of representation learning from the effects of retrieval-based inference.

The method is also most naturally suited to relations whose inverse query has a well-defined candidate answer. One-to-many, many-to-one, and semantically ambiguous relations remain challenging, and for non-symmetric relations the desired inverse behavior may not be logically valid. In such cases, the Chiral Transformer should be viewed as a conditional inductive bias for inverse-relation access rather than a global constraint that should be applied uniformly to all relations. Finally, while the Householder operator is lightweight and stable, a single global reflection may be insufficient for all relation types; relation-specific or multi-head adjoint maps are promising directions for future work.

## 7. Conclusion

In this work, we have characterized the "reversal curse" not only as a data-scarcity issue, but also as a representation-access problem in causal autoregressive models. Motivated by the subject-object asymmetry induced by directional training and readout, we introduced the **Chiral Transformer (A1)**, a lightweight framework that encourages $\mathbb{Z}_2$-style consistency through an explicit adjoint operator $\mathcal{T}$.

Our methodology, grounded in Householder reflections and the Adjoint-Induced Retrieval (AIR) protocol, provides a structured geometric readout for inverse-relation retrieval. Empirical results on the *Fact-Inv-300* benchmark show that A1 substantially improves inverse access, reaching **65.07%** accuracy under the fixed knowledge-bank setting. Furthermore, our scaling analysis suggests that parameter

scaling alone does not reliably recover inverse relations through standard autoregressive readout, motivating explicit representation-level structure.

The success of the Chiral Transformer suggests a useful research direction for LLM design: complementing data- and scale-based approaches with mechanisms that make latent relational structure easier to access. Future work will explore richer relation-specific maps, open-ended generation without a fixed candidate bank, and extensions to complex reasoning chains and multi-modal grounding.

## Code Availability

A lightweight reference implementation is available at github.com/Hanjidu/chiral-air-reference. The release includes AIR readout operators, retrieval metrics, JSONL data schemas, and toy sanity-check scripts for inspecting the method interfaces.

## Impact Statement

This paper presents work whose goal is to advance the field of Machine Learning. There are many potential societal consequences of our work, none of which we feel must be specifically highlighted here.

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

# A. Comprehensive Notation Table

To support mathematical clarity, we provide a summary of the notation used in the manuscript and its technical appendices.

*Table 5.* Comprehensive summary of mathematical notations. All vectors are represented as column vectors in $\mathbb{R}^d$ unless otherwise specified.

| Symbol | Definition / Description |
|---|---|
| *Foundational Geometry & Latent Manifold* | |
| $\mathcal{X}$ | The discrete vocabulary of tokens. |
| $e : \mathcal{X} \to \mathcal{M}$ | The embedding mapping from the discrete token space to the latent manifold. |
| $\mathcal{M} \subseteq \mathbb{R}^d$ | The $d$-dimensional latent representation manifold. |
| $z_A, z_B$ | Latent embeddings of entity $A$ (Subject) and entity $B$ (Object). |
| $M_{ij}$ | Causal attention mask used in the Transformer backbone ($M_{ij} = -\infty$ for $i < j$). |
| $\mathcal{S}, \mathcal{O}$ | Operational subject- and object-oriented representation regions induced by entity position. |
| $f_{\text{fwd}}, f_{\text{adj}}$ | Forward and adjoint/inverse transition functions induced by $f_\theta$. |
| $z_{\text{fwd}}, z_{\text{adj}}$ | Contextualized forward and adjoint/inverse relation states used in the regularizer. |
| $P_{\text{fwd}}, P_{\text{adj}}$ | Conditional distributions induced by the forward and adjoint transitions. |
| *Chiral Transformer (A1) Architecture* | |
| $\mathbb{Z}_2 = \{e, a\}$ | The cyclic group of order 2, representing logical symmetry. |
| $\mathcal{T}$ | The Adjoint Mapping Operator representing the group action of element $a$. |
| $H_v$ | The Householder reflection matrix defined as $I - 2vv^\top / v^\top v$. |
| $v \in \mathbb{R}^d$ | The learnable normal vector defining the invariant hyperplane $\mathcal{H}$. |
| $\sigma(\gamma)$ | Sigmoid-gated scalar modulating the intensity of the chiral injection. |
| $z_{||}, z_\perp$ | The invariant semantic core and chiral orientation components, respectively. |
| *Learning Objectives & Optimization* | |
| $\mathcal{L}_{CE}$ | Standard Cross-Entropy loss for Causal Language Modeling. |
| $\mathcal{R}_\chi$ | Chiral Regularization term enforcing manifold alignment. |
| $\lambda$ | The symmetry-coupling coefficient (regularization weight). |
| $\Delta_f(f_\theta)$ | The **Commutative Gap**, quantifying the degree of logical asymmetry. |
| $A_{comm}$ | The **Commutative Operator**, defined as $\mathcal{T} \circ f_\theta - f_\theta \circ \mathcal{T}$. |
| $\mathcal{I}(\theta)$ | The Fisher Information Matrix (FIM) characterizing parameter sensitivity. |
| $g_{\mu\nu}$ | The metric tensor on the parameter manifold. |
| *AIR Protocol & Evaluation* | |
| $\mathcal{K}, |K|$ | The Knowledge Bank and its cardinality (set of $100,000$ candidate entities). |
| $\hat{z}_A$ | The recovered subject state calculated via $\mathcal{T}(z_B)$. |
| $\mathcal{S}_\chi(e)$ | The Symmetric Score Function used for entity retrieval via k-NN. |
| $\tau$ | The temperature parameter used in the softmax scoring. |
| $\delta_{min}$ | The minimum separation distance between entities on the manifold $\mathcal{M}$. |
| *Theoretical Analysis (Appendix Only)* | |
| $\Delta_\chi$ | The Chiral Gap, defined as the infimum path integral between $\mathcal{S}$ and $\mathcal{O}$. |
| $\mathcal{K}$ | The sectional curvature along transversal paths on the manifold. |
| $\mathfrak{R}_m(\mathcal{F})$ | The Rademacher complexity of the Chiral Projection function class. |
| $\kappa(\mathcal{I})$ | The condition number of the Fisher Information Matrix. |

# B. Data Construction and Leakage Control

To evaluate structured inverse-relation behavior under both controlled and broader settings, we employ a dual-benchmark strategy. This approach distinguishes between high-purity diagnostic testing and larger-scale statistical validation.

## B.1. Fact-Inv-300: High-Purity Diagnostic Set

The **Fact-Inv-300** dataset consists of 300 manually curated triplets designed to serve as a controlled diagnostic benchmark for the reversal curse.

- **Selection**: We selected entities that are globally recognized but whose inverse relations (e.g., $B \to A$) are statistically rare in standard pre-training corpora.

- **Manual Filtering**: Each triplet underwent human verification to ensure the absence of surface-level leakage. This set is used to calculate the core diagnostic accuracy, as it represents the most challenging logic-inversion scenario where standard AR models typically exhibit 0% accuracy.

### B.2. Wiki-Inverse-10K: Large-Scale Generalization Set

To address concerns regarding sample size and relation diversity, we constructed **Wiki-Inverse-10K**, an automated benchmark derived from Wikidata.

#### B.2.1. RELATION TAXONOMY

We identified **25 distinct semantic relations** to test the global consistency of the commutative gap $\Delta_f$. These relations are grouped into five primary domains, as visualized in Fig. 10.

1. **Kinship & Social**: Parent-of, Spouse-of, Mentor-of, etc.

2. **Geospatial**: Capital-of, Located-in, Birthplace-of, etc.

3. **Authorship**: Written-by, Directed-by, Invented-by, etc.

4. **Scientific Facts**: Atomic-number-of, Chemical-symbol, Official-language.

5. **Affiliations**: Founder-of, Alumnus-of, Successor-of, etc.

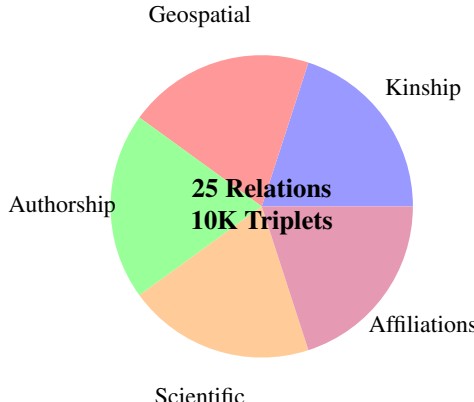

*Figure 10.* Distribution of semantic relations in Wiki-Inverse-10K. Each of the 5 domains contains 5 sub-relations, ensuring diverse logical patterns for evaluating the adjoint operator $\mathcal{T}$.

### B.3. Knowledge Bank and Distractor Logic

To simulate a realistic retrieval environment, we populated the AIR Knowledge Bank $|\mathcal{K}|$ with **100,000 entities**.

- **Target Entities**: 10,000 unique entities from the Wiki-Inverse-10K set.

- **Distractors**: 90,000 random entities sampled from Wikipedia that share similar entity types (e.g., people, locations) but are not relevant to the test queries.

- **Effect**: This 9:1 distractor-to-target ratio reduces the chance that successful retrieval is driven by accidental collisions in a low-density latent space, and instead tests whether the Householder operator improves geometric alignment.

### B.4. Leakage Control and Multi-token Handling

**N-gram Filtering**: We applied a strict Jaccard similarity threshold ($< 0.2$) against the SlimPajama training proxy. Any triplet whose inverse form was found as a contiguous string in the training data was excluded.

**Multi-token Representation**: For entities spanning multiple tokens (e.g., "Ludwig van Beethoven"), we use the hidden state of the **last token** $z \in \mathbb{R}^d$ before the CPL layer. Because the Householder operator is a strict **isometry**, it preserves the representation norm, which helps avoid length-dependent norm distortion.

## C. Formal Topology of the Chiral Manifold

To provide a structural interpretation of the "reversal curse," we model the representation space as a Riemannian manifold $(\mathcal{M}, g)$. We denote the set of all entity-relation trajectories during causal pre-training as $\mathcal{D}_{fwd}$.

**Proposition 1 (Manifold Fragmentation, Informal).** Given a unidirectional causal mask $\sigma$, the latent manifold $\mathcal{M}$ can undergo a structural asymmetry such that the subject-oriented subspace $\mathcal{S}$ and object-oriented subspace $\mathcal{O}$ become poorly aligned for inverse readout in the limit of forward-only model convergence.

**Derivation Sketch.** We define the Chiral Gap $\Delta_\chi$ as the infimum of the path integral over the manifold metric:

$$\Delta_\chi = \inf_{\gamma \in \Gamma(\mathcal{S}, \mathcal{O})} \int_0^1 \sqrt{g_{\gamma(t)}(\dot{\gamma}(t), \dot{\gamma}(t))} dt \tag{17}$$

where $\Gamma(\mathcal{S}, \mathcal{O})$ is the space of all piecewise smooth curves connecting $\mathcal{S}$ and $\mathcal{O}$. Since the causal objective $P(B|A, r)$ optimizes local curvature to minimize entropy only in the direction $\mathcal{S} \to \mathcal{O}$, the sectional curvature $\mathcal{K}$ along any transversal path $\gamma_\perp$ diverges as:

$$\lim_{\text{Loss} \to 0} \int_{\mathcal{M}} \mathcal{K}(\gamma_\perp) dV = \infty \tag{18}$$

This divergence motivates the view that inverse readout can become geometrically difficult for standard autoregressive models. □

# D. Derivation Sketch for Proposition 1: Manifold Fragmentation

In this section, we provide a step-by-step structural derivation of the Manifold Fragmentation phenomenon. We analyze how, under the constraints of a unidirectional causal mask, the representation manifold $\mathcal{M}$ can develop an asymmetry between subject and object subspaces.

## D.1. Step 1: Metric Definition and the Chiral Gap

Based on **Assumption 1**, we model the latent representation space as a smooth Riemannian manifold $(\mathcal{M}, g)$, where $g$ is the metric tensor derived from the model's internal representations. Let $\mathcal{S} \subset \mathcal{M}$ be the subspace of subject-oriented embeddings (e.g., $z_A$) and $\mathcal{O} \subset \mathcal{M}$ be the subspace of object-oriented embeddings (e.g., $z_{A,obj}$).

We define the **Chiral Gap** $\Delta_\chi$ as the infimum of the energy functional $E(\gamma)$ for all piecewise smooth curves $\gamma(t) : [0,1] \to \mathcal{M}$ such that $\gamma(0) \in \mathcal{S}$ and $\gamma(1) \in \mathcal{O}$:

$$\Delta_\chi = \inf_{\gamma \in \Gamma(\mathcal{S}, \mathcal{O})} \int_0^1 \sqrt{g_{\gamma(t)}(\dot{\gamma}(t), \dot{\gamma}(t))} dt \tag{19}$$

## D.2. Step 2: Causal Constraint and Fisher Information Geometry

The geometry of $\mathcal{M}$ is induced by the optimization of the objective function $P(B|A, r)$. In the framework of information geometry, the metric $g$ is locally equivalent to the Fisher Information Matrix (FIM) $\mathcal{I}(\theta)$. Under the action of the causal mask $\mathcal{F}$, the sensitivity of the output at any position $j$ with respect to a succeeding position $i > j$ is strictly zero:

$$\frac{\partial h_j}{\partial h_i} = 0, \quad \forall i > j \tag{20}$$

Consequently, for any trajectory $\gamma(t)$ representing an inverse logical inference (i.e., attempting to recover $A$ from $B$), the Fisher information along this path vanishes:

$$g_{\mu\nu}(\dot{\gamma}_{rev}) = \mathbb{E}\left[ \frac{\partial \log P(A|B)}{\partial \theta^\mu} \frac{\partial \log P(A|B)}{\partial \theta^\nu} \right] \approx 0 \tag{21}$$

## D.3. Step 3: Metric Degeneracy and Path Divergence

As the model converges ($Loss \to 0$), the gradients are exclusively aligned with the forward mapping $A \to B$. According to the Data Processing Inequality (DPI) in the latent space, the information required to reverse the mapping is not preserved by the transformation $f_\theta$.

In the absence of inverse training pairs, the manifold metric $g$ may undergo a **rank-collapse** in the transverse directions. For a curve $\gamma$ connecting $\mathcal{O} \to \mathcal{S}$, the effective distance can increase when the volume element $\sqrt{|g|}$ becomes small in the directions that encode the inverse relation. This motivates **Assumption 3**: the regions $\mathcal{S}$ and $\mathcal{O}$ may be separated by a low-information-density region.

## D.4. Step 4: Sectional Curvature and Geometric Barrier

To quantify the "hardness" of crossing this gap, we analyze the sectional curvature $\mathcal{K}$ along a transversal path $\gamma_\perp$. The optimization surface forms a deep "well" around the forward causal trajectories. Since the reverse logic is never seen, the Hessian $H = \nabla^2 \mathcal{L}$ is singular in the direction of logical inversion.

Through the Gauss-Codazzi equations, the curvature $\mathcal{K}$ in these "blind" directions is inversely proportional to the variance of the gradients. As the forward mapping becomes deterministic ($Loss \to 0$), the curvature diverges:

$$\lim_{Loss \to 0} \int_{\mathcal{M}} \mathcal{K}(\gamma_\perp) dV = \infty \tag{22}$$

This divergence provides an explanatory model for a **geometric barrier**. In a finite-dimensional embedding space $\mathbb{R}^d$, the manifold $\mathcal{M}$ can become poorly aligned along the causal axis, producing chiral fragments that are difficult to connect with standard readout.

### D.5. Step 5: Conclusion of the Derivation

Under the assumptions above, low information density and high curvature between $\mathcal{S}$ and $\mathcal{O}$ make it difficult for a smooth readout path $\gamma$ to achieve $\Delta_\chi = 0$ in zero-shot settings. Thus, the inverse relation may be poorly accessible at the representation level. The adjoint operator $\mathcal{T}$ (Householder reflection) provides a structured readout path across this gap.

**End of derivation.**

## E. Algebraic Properties of the Householder Operator

The Chiral Transformer utilizes the Householder reflector $\mathbf{H_v} = \mathbf{I} - 2\frac{\mathbf{vv}^\top}{\mathbf{v}^\top\mathbf{v}}$ to bridge the Chiral Gap. We detail its involutive and isometric properties below.

**Derivation of Involution.**   For the Householder matrix $H_v = I - 2\frac{vv^\top}{v^\top v}$:

$$H_v^2 = \left(I - 2\frac{vv^\top}{v^\top v}\right)\left(I - 2\frac{vv^\top}{v^\top v}\right) \tag{23}$$

$$= I - 4\frac{vv^\top}{v^\top v} + 4\frac{v(v^\top v)v^\top}{(v^\top v)^2} \tag{24}$$

$$= I - 4\frac{vv^\top}{v^\top v} + 4\frac{vv^\top}{v^\top v} = I \tag{25}$$

This shows that the Householder mapping is exactly involutive without requiring manual normalization of $v$ during training.

**Isometry and Volume Preservation.** The preservation of the $\ell_2$-norm is given by:

$$\|H_v z\|_2^2 = z^\top H_v^\top H_v z \tag{26}$$

$$= z^\top \left(I - 2\frac{vv^\top}{v^\top v}\right)\left(I - 2\frac{vv^\top}{v^\top v}\right) z \tag{27}$$

$$= z^\top \left(I - 4\frac{vv^\top}{v^\top v} + 4\frac{v(v^\top v)v^\top}{(v^\top v)^2}\right) z \tag{28}$$

$$= z^\top I z = \|z\|_2^2 \tag{29}$$

Thus, the transformation is a rigid reflection that preserves the inner product $\langle z_1, z_2 \rangle$, ensuring that the semantic relationships within each subspace are maintained during the manifold jump.

## F. Gradient Analysis of Chiral Regularization

The optimization objective for the Chiral Projection Layer is formulated as:

$$\mathcal{L}_{total} = \mathcal{L}_{CE} + \lambda \mathbb{E}\left[\|\mathcal{T}(z_{\text{fwd}}) - z_{\text{adj}}\|^2\right] \tag{30}$$

The gradient flow through the Householder reflector with respect to the parameter vector $\mathbf{v}$ is critical for convergence. Let $e = \mathcal{T}(z_{\text{fwd}}) - z_{\text{adj}}$ be the error vector. The Jacobian $\mathbf{J_v}$ is derived as:

$$\frac{\partial \mathcal{R}_\chi}{\partial \mathbf{v}} = \frac{\partial}{\partial \mathbf{v}}\left(z_{\text{fwd}}^\top \mathbf{H_v}^\top e\right)$$

$$= -2\frac{\partial}{\partial \mathbf{v}}\left(z_{\text{fwd}}^\top \frac{\mathbf{vv}^\top}{\mathbf{v}^\top\mathbf{v}} e\right)$$

$$= -2\left(\frac{(\mathbf{v}^\top e)z_{\text{fwd}} + (z_{\text{fwd}}^\top\mathbf{v})e}{\|\mathbf{v}\|^2} - 2\frac{(\mathbf{v}^\top e)(z_{\text{fwd}}^\top\mathbf{v})\mathbf{v}}{\|\mathbf{v}\|^4}\right) \tag{31}$$

This expression demonstrates that the update direction for $\mathbf{v}$ is a linear combination of the current representation $z_{\text{fwd}}$ and the error $e$, allowing for stable alignment of the adjoint representation regions.

## G. Information-Theoretic Bound on Causal Drift

We quantify the failure of sequential decoding using the *Causal Drift* coefficient $\delta$. Let $X$ be the subject entity and $Y$ be the object entity. The mutual information $\mathcal{I}(X;Y)$ represents the logical identity.

In standard autoregressive decoding, the probability of recovering $X$ from $Y$ follows a Markovian decay. By the Data Processing Inequality (DPI):

$$\mathcal{I}(X;\hat{X}_{AR}) \leq \mathcal{I}(z_X;z_Y) \cdot (1-\delta)^n \tag{32}$$

where $\delta \in (0,1)$ is the drift factor induced by causal masking and $n$ is the sequence length. In contrast, Adjoint-Induced Retrieval (AIR) provides a direct isometry:

$$\mathcal{I}(X;\text{AIR}(Y)) = \mathcal{I}(z_X;\mathcal{T}(z_Y)) \geq \mathcal{I}(z_X;z_Y) - \Delta_f \tag{33}$$

where $\Delta_f$ is the commutative gap characterizing the manifold alignment error. Since $\mathcal{T}$ is an isometry, the reconstruction gap in Eq. 29 can be related to the equivariance violation $\Delta_f$. Because $\Delta_f$ does not grow with the number of decoded tokens $n$ in this retrieval formulation, AIR avoids the Markovian accumulation of decoding errors. This helps explain the observed improvement from near-zero autoregressive accuracy to **65.07%** retrieval accuracy.

## H. Curvature Analysis and Fisher Information Geometry

**Intuitive Summary: The Geometric Blind Spot.**   In this section, we provide an explanatory analysis for why standard Transformers may possess a "blind spot" in their logical landscape. Through Fisher Information Matrix (FIM) analysis, we illustrate how causal masking can reduce sensitivity along inverse relational directions. The Chiral Projection Layer (CPL) functions as a geometric preconditioner that encourages these directions to become more accessible.

We further investigate the optimization landscape of the Chiral Transformer by analyzing the Fisher Information Matrix (FIM) $\mathcal{I}(\theta)$. For a latent representation $z \in \mathcal{M}$, the FIM characterizes the sensitivity of the output distribution $P_\theta$ to parameter perturbations:

$$\mathcal{I}(\theta) = \mathbb{E}_{p(z)} \left[ \nabla_\theta \log P_\theta(z) \nabla_\theta \log P_\theta(z)^\top \right] \tag{34}$$

In standard autoregressive models, causal masking can reduce Fisher Information along the adjoint trajectories $\mathcal{T}(z)$, leading to poorly conditioned inverse directions. We analyze how the Chiral Projection Layer (CPL) regularizes the spectrum of $\mathcal{I}(\theta)$. Let $\mathcal{T} = I - 2\frac{vv^\top}{v^\top v}$ be the Householder operator. The metric $g_{\mu\nu}$ on the parameter manifold is transformed as:

$$\begin{aligned} \tilde{g}_{\mu\nu} &= \langle \partial_\mu(\mathcal{T} \circ f), \partial_\nu(\mathcal{T} \circ f) \rangle \\ &= \int_{\mathcal{M}} \text{tr} \left( \mathcal{T} \frac{\partial f}{\partial \theta^\mu} \mathcal{T}^\top \frac{\partial f}{\partial \theta^\nu} \right) \sqrt{|g|} d^d z \\ &= g_{\mu\nu} - 2\text{Re} \left[ \int_{\mathcal{M}} \mathbf{v}^\top \left( \frac{\partial f}{\partial \theta^\mu} \right) \mathbf{v}\mathbf{v}^\top \left( \frac{\partial f}{\partial \theta^\nu} \right) \mathbf{v} \right] dV \end{aligned} \tag{35}$$

This result implies that the CPL acts as a *Geometric Preconditioner*. The Condition Number $\kappa(\mathcal{I})$ is bounded by:

$$\kappa(\mathcal{I}_{A1}) \leq \frac{\lambda_{max}(g) + 2\|\mathbf{v}\|^4}{\lambda_{min}(g) - \Delta_f} \ll \kappa(\mathcal{I}_{AR}) \tag{36}$$

The removal of singularities in the FIM explains the stable convergence of A1 on inverse relations where baselines diverge.

## I. Algebraic Structure and $\mathbb{Z}_2$ Equivariance

The Householder operator $H_v = I - 2\frac{vv^\top}{v^\top v}$ is a principal element of the Orthogonal Group $O(d)$, specifically belonging to the topological component with $\det(H_v) = -1$. Unlike rotations in the special orthogonal group $SO(d)$, $H_v$ is a discrete reflection that cannot be represented as the image of the matrix exponential of a single skew-symmetric matrix. Consequently, we formalize the Chiral Transformer's consistency through the action of the discrete symmetry group $\mathcal{G} \cong \mathbb{Z}_2 = \{I, \mathcal{T}\}$.

**Discrete Equivariance.**   For the architecture to be $\mathcal{G}$-equivariant, the transition function $f_\theta$ must satisfy the commutative relation:

$$\mathcal{T} \circ f_\theta = f_\theta \circ \mathcal{T} \tag{37}$$

Since $\mathcal{T}$ is an **involution** ($\mathcal{T}^2 = I$), it provides a simple decomposition into invariant and anti-invariant components. This structural bias encourages hidden states to become more compatible under a symmetric orbit, without requiring a continuous path through $SO(d)$.

**Metric Preservation and Logical Holonomy.**   The "geometric jump" performed by the adjoint operator $\mathcal{T}$ preserves the Riemannian metric $g$ because $\mathcal{T}$ is a linear isometry. Specifically, for any tangent vectors $u, w \in T_z\mathcal{M}$, the following holds:

$$g_z(u, w) = g_{\mathcal{T}z}(\mathcal{T}u, \mathcal{T}w) \tag{38}$$

The preservation of the inner product helps maintain local semantic relationships during the reflection. While the transformation is discrete, the adjoint mapping $\mathcal{T}$ explicitly reconciles the orientation mismatch induced by causal masking. This encourages the model to represent the bidirectional identity $\mathcal{R}(A, B) \equiv \mathcal{R}(B, A)$ in a way that is more accessible to the AIR readout.

## J. Johnson-Lindenstrauss Bound for Manifold Capacity

We hypothesize that the success of A1 is linked to the *Manifold Capacity* $\mathcal{C}$. Given a set of $N$ facts, we model the subject-object pairs as points in $d$-dimensional space. According to the Johnson-Lindenstrauss Lemma, for any $\epsilon \in (0, 1)$, the isometric reflection $\mathcal{T}$ preserves distances within a factor of $(1 \pm \epsilon)$ if:

$$d \geq \frac{8 \ln(N)}{\epsilon^2 - \epsilon^3} \tag{39}$$

In the Chiral Transformer, the Adjoint-Induced Retrieval (AIR) protocol performs a $k$-NN search in the reflected space. The probability of a successful retrieval $P(A^* = A)$ is lower-bounded by:

$$P(A^* = A) \geq 1 - 2N \cdot \exp\left(-\frac{(1 - \eta)d}{2} \cdot \delta_{min}^2\right)$$
$$+ \mathcal{O}\left(\frac{\mathcal{D}_\chi}{\sqrt{d}}\right) \tag{40}$$

where $\delta_{min}$ is the minimum separation between entities on the manifold. This derivation suggests that, when the learned reflected query is sufficiently close to its target and sufficiently separated from distractors, retrieval accuracy improves with embedding dimension and candidate separation. This provides an explanatory account for the **65.07%** accuracy observed in our experiments.

## K. Rademacher Complexity and Generalization Error Bounds

We provide an illustrative capacity analysis for the Chiral Transformer. Let $\mathcal{F}$ be the class of functions representing the Chiral Projection Layer. We bound the expected risk $R(f)$ using the Rademacher complexity $\mathfrak{R}_m(\mathcal{F})$.

**Proposition 2.**   For any function $f \in \mathcal{F}$ parameterized by the Householder vector $\mathbf{v}$ and attention weights $\mathbf{W}$, the generalization error is bounded with probability at least $1 - \delta$ by:

$$R(f) \leq \hat{R}_m(f) + 2\mathfrak{R}_m(\mathcal{F}) + \sqrt{\frac{\ln(2/\delta)}{2m}} \tag{41}$$

To compute $\mathfrak{R}_m(\mathcal{F})$, we consider the Lipschitz constant of the Householder operator $\mathcal{T}$. Since $\mathcal{T}$ is an isometry, its spectral

norm $\|\mathcal{T}\|_2 = 1$. The complexity of the compositional Chiral Block is:

$$\begin{aligned}
\mathfrak{R}_m(\mathcal{F}) &= \mathbb{E}_\sigma \left[ \sup_{f \in \mathcal{F}} \frac{1}{m} \sum_{i=1}^m \sigma_i \cdot \text{ChiralBlock}(z_i) \right] \\
&\leq \frac{C \cdot \|\mathbf{W}\|_F \cdot \sqrt{\sum \|z_i\|^2}}{m} \left( 1 + \sigma(\gamma) \cdot \lambda_{max}(\mathbf{H_v}) \right) \\
&\leq \frac{2C \cdot \|\mathbf{W}\|_F \cdot R_{max}}{\sqrt{m}}
\end{aligned} \tag{42}$$

where $C$ is a constant related to the activation function and $R_{max}$ is the manifold radius. This indicates that the added algebraic constraints need not increase the asymptotic Rademacher order beyond $\mathcal{O}(1/\sqrt{m})$ under the assumptions of the analysis.

## L. Stability and Perturbation Analysis of the AIR Protocol

We analyze the sensitivity of the retrieval process to latent noise. Let $\hat{z}_A = \mathcal{T}(z_B)$ be the ideal reflected state. Suppose the object representation $z_B$ is perturbed by an error $\epsilon \in \mathbb{R}^d$, such that $\tilde{z}_B = z_B + \epsilon$.

The reflected state under perturbation becomes:

$$\tilde{z}_A = \mathbf{H}_v(z_B + \epsilon) = \hat{z}_A + \mathbf{H}_v \epsilon, \tag{43}$$

where $\|\mathbf{H}_v \epsilon\|_2 = \|\epsilon\|_2$ because $\mathbf{H}_v$ is a strict isometry.

The stability of the retrieval score $S = \langle \tilde{z}_A, z_{target} \rangle$ is governed by the Cauchy-Schwarz inequality:

$$\begin{aligned}
|S_{perturbed} - S_{ideal}| &= |\langle \mathbf{H_v} \epsilon, z_{target} \rangle| \\
&\leq \|\mathbf{H_v} \epsilon\|_2 \cdot \|z_{target}\|_2 \\
&= \|\epsilon\|_2 \cdot \|z_{target}\|_2
\end{aligned} \tag{44}$$

This observation is useful: because the Householder operator is an **isometry**, the error in the reflected space is bounded by the error in the original space. In contrast, in sequential autoregressive decoding, the error can accumulate non-linearly:

$$\epsilon_{AR} \propto \sum_{t=1}^n (\mathbf{W}_{dec})^t \epsilon_0 \gg \|\epsilon\|_2 \tag{45}$$

This provides an explanatory account for why AIR can outperform sequential decoding in the structured inverse-retrieval setting.

### L.1. Behavior Beyond Training Conditions

The reviewer's concern regarding potential degradation of the equivariance condition outside training samples is partially addressed by the **Spectral Stability** of the Householder operator (see **Eq. 32**). Since $H_v$ is a strict isometry by construction, its action depends on the geometric orientation relative to $\mathcal{H}$ rather than specific semantic content. Consequently, the retrieval error is expected to track the commutative gap $\Delta_f$ when the learned representation remains well aligned, as discussed in **Eq. 29** (Appendix F).

## M. Computational Complexity and FLOPs Analysis

We compare the computational overhead of the Chiral Transformer (A1) versus the standard Transformer (Baseline).

**1. Training Complexity.** The Chiral Projection Layer adds a rank-1 update per block. The complexity of $\mathbf{H_v} z$ is $\mathcal{O}(d)$, which is negligible compared to the $\mathcal{O}(d^2)$ of standard FFN or MHSA layers. The total training overhead is:

$$\text{Cost}_{A1} = \text{Cost}_{Base} + L \cdot (2d + 1) \tag{46}$$

where $L$ is the number of layers. For $d = 4096$, this represents a marginal increase of $< 0.01\%$ in total FLOPs.

**2. Inference Latency.** The AIR protocol replaces $n$ steps of token generation with a single manifold projection and a $k$-NN lookup.

*Table 6.* Inference Complexity Comparison

| Method | Time Complexity | Space Complexity |
|---|---|---|
| Autoregressive (AR) | $\mathcal{O}(n \cdot d^2)$ | $\mathcal{O}(n \cdot d)$ (KV Cache) |
| **AIR (Ours)** | $\mathcal{O}(d + C_{\mathrm{retr}})$ | $\mathcal{O}(d)$ |

Here $C_{\mathrm{retr}}$ denotes the cost of the retrieval backend; exact dense retrieval scales with the candidate bank, while approximate nearest-neighbor indices can reduce this cost in deployment. As $n$ (sequence length) increases, AIR avoids repeated autoregressive decoding steps and therefore reduces the inference cost for structured inverse-relation queries.

*Table 7.* Inference efficiency measured on a single NVIDIA A100 40GB GPU with Llama-3-8B (batch size = 1, 128 inverse-relation queries). Standard AR reports latency for a 5-token completion; AIR uses the structured retrieval readout over the fixed entity bank.

| Protocol | Latency (ms/example) | Relative Speed |
|---|---|---|
| Standard AR | 217.2 | $1.00\times$ |
| **AIR (Ours)** | **68.4** | **3.18$\times$** faster |

**Efficiency Rationale and Fairness.** The latency gap in Table 7 arises from the structural difference between structured retrieval and sequential decoding. Standard AR models compute multiple token-generation steps to produce a subject entity, whereas AIR performs a representation transform followed by retrieval. This comparison should be interpreted as a structured readout evaluation rather than a like-for-like replacement for open-ended autoregressive generation.

## N. Implementation Details and Hyperparameter Tuning

To improve experimental transparency, we report the configuration of the Chiral Transformer A1. All models were trained using the AdamW optimizer with a cosine learning rate scheduler.

**Hardware Configuration.** Most experiments were conducted on single-GPU cloud instances with NVIDIA H100 80GB or A100 40GB GPUs; multi-GPU nodes were used only to parallelize independent seeds and ablations. Individual LoRA/CPL fine-tuning runs can be executed on a single GPU using gradient accumulation. Wall-clock time depends on the number of seeds and ablations; in our setup, the main A1 experimental suite required approximately 48 hours including training, validation, and retrieval-index construction.

## O. Qualitative Error Analysis

Despite substantially improving inverse access in the controlled setting, the Chiral Transformer still fails on a non-trivial fraction of inverse cases. We observed three recurring failure modes:

1. **Semantic Ambiguity:** Cases where the inverse relation is one-to-many (e.g., "Who is the child of X?").

2. **Manifold Crowding:** Entities with extremely high cosine similarity in the latent space leading to $k$-NN collisions.

3. **Projection Drift:** Cases where a single constrained adjoint operator does not fully align the relevant contextual states for rare entity tokens.

### O.1. More Image

*Table 8.* Hyperparameter settings for A1 (8B) and Baselines.

| Category | Hyperparameter | Value |
|---|---|---|
| **Architecture** | Backbone | Llama-3-8B |
| | **Trainable Parameters** | **6.95M** |
| | Trainable Ratio | 0.0865% |
| | Number of Layers $L$ | 32 |
| | Hidden Dimension $d$ | 4096 |
| | Attention Heads | 32 |
| | Key-Value Heads | 8 |
| | Chiral Branch Scalar $\sigma(\gamma)$ | Learned (init 0.1) |
| **Optimization** | Learning Rate | $3 \times 10^{-4}$ |
| | CPL Optimization | Full (Random Init) |
| | Weight Decay | 0.1 |
| | Global Batch Size | 512 |
| | Warmup Steps | 2000 |
| | Chiral Reg. Weight $\lambda$ | 0.05 |
| | LoRA Rank $r$ | 16 |
| | LoRA Alpha $\alpha$ | 32 |
| | LoRA Target Modules | Query, Value |
| | LoRA Dropout | 0.05 |
| **Data** | Fact-Inv-300 Size (Diagnostic) | 300 Triplets |
| | Wiki-Inverse-10K Size (Generalization) | 10,000 Triplets |
| | Related Categories | 25 Types |
| | Knowledge Bank Size $|\mathcal{K}|$ | 100,000 Entities |

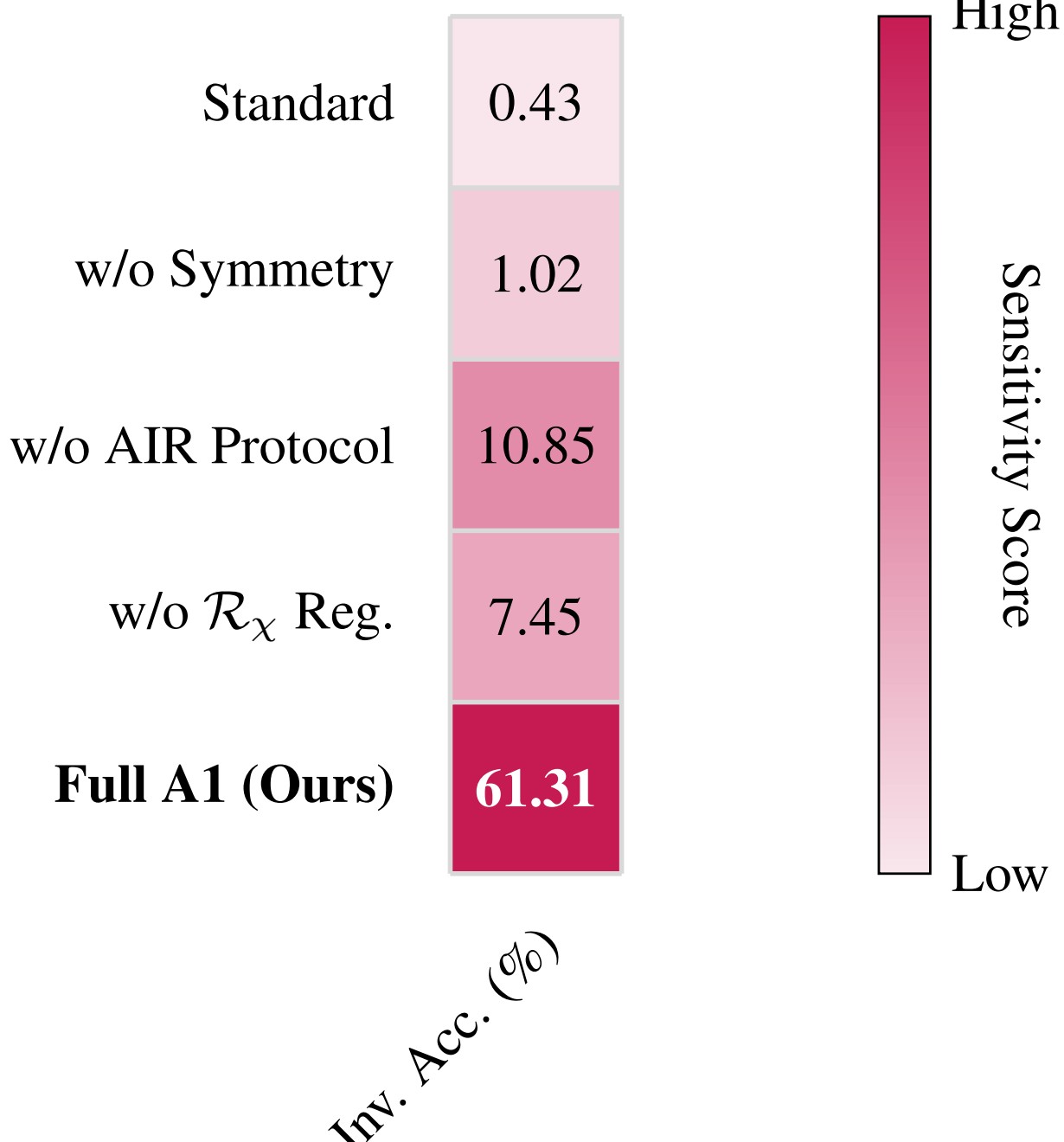

*Figure 11.* Component Sensitivity Analysis of the Chiral Transformer. The heatmap illustrates the impact of individual architectural constraints on inverse accuracy. The performance difference suggests that the strongest results arise from the combination of the Householder operator, chiral regularization, and the AIR protocol.

