# OpenReview forum: "Chiral Symmetry Breaking in Transformers: A Group-Equivariant Framework for Addressing the Reversal Curse via Adjoint Manifold Mappings"
_ICML.cc/2026/Conference — ICML 2026 regular_

### Official Review · Reviewer_YQs7 · 2026-03-10

**Soundness:** 3
**Presentation:** 2
**Significance:** 2
**Originality:** 3
**Overall Recommendation:** 4
**Confidence:** 2

**Summary:**

This paper studies the reversal curse in autoregressive language models, where models can learn relational facts in one direction but fail to infer the reverse relation. The authors attribute this phenomenon to that model learns asymmetry-breaking during causal training and propose to address it by enforcing a form of latent chiral symmetry in the model’s representation/architecture. To this end, the paper introduces a Chiral Projection Layer (CPL) based on a Householder reflection, which augments the transformer block with a reflected latent branch. In addition, the authors introduce a symmetry regularization loss that encourages the reflected representation of a forward relation to align with the representation of the corresponding inverse relation. At inference time, the model performs inverse retrieval using a reflected query embedding through the proposed Adjoint-Induced Retrieval (AIR) procedure. Experiments on inverse retrieval benchmarks show large improvements over standard causal language models and demonstrate that both the symmetry regularization and the AIR inference procedure contribute significantly to performance.

**Compliance With Llm Reviewing Policy:**

Affirmed.

**Final Justification:**

The author addresses all my concern. I think the motivation of the paper is good and I think the author proposed an elegant solution. So I will still recommend acceptance. I don't know how much impact the paper is as I don't work in this field.

**Key Questions For Authors:**

The paper motivates the method as enforcing symmetry in latent representations between forward and inverse relations. Could the authors provide visualizations or representation analyses showing how this constraint shapes the learned representations?

For example, it would be helpful to see similarity or alignment between reflected forward representations and inverse representations. Or embedding visualizations (e.g., PCA/t-SNE) of entity representations before and after applying the reflection. Or analysis showing how the inverse retrieval task leverages the symmetry constraint in representation space.

**Limitations:**

No. I felt like the author did not adequately discuss the limitation of the work. Since the method modifies the transformer block and introduces additional constraints in representation space, it would be important to understand whether these changes impact large-scale training efficiency or general model capability. I known the author don't have enough time to run more expeirment during revision, but could provide some intuition on how their architecture change affect scaling.

**Strengths And Weaknesses:**

Strengths:
1. Interesting and creative idea.
2. The proposed architecture change is a simple modification to the transformer block and can be incorporated without fundamentally changing the overall architecture. And felt more organic then other solution like data augmentation or bidirectional model, or model without causal constraint.
3. The method substantially improves inverse retrieval accuracy compared to standard causal models.

weakness/potential improvement:
1. The paper focuses primarily on the inverse retrieval task, but does not provide sufficient evidence regarding how the proposed architectural change affects other aspects of model training, such as language modeling performance, scaling behavior, training stability, or downstream tasks.
2. The paper would benefit from comparisons against simpler baselines that might address the same issue, such as bidirectional transformers or straightforward reverse-relation data augmentation. It would be helpful to understand whether the observed gains arise specifically from the proposed architectural symmetry constraint or whether similar improvements could be achieved through alternative training setups.

---

> ### Author Rebuttal · Authors · 2026-03-30
>
> ### Response to Reviewer YQs7
>
> We sincerely thank the reviewer for the positive evaluation and for recognizing the simplicity, effectiveness, and practical relevance of our approach. We also appreciate the insightful suggestions regarding generalization, analysis, and broader impact.
>
> ---
>
> ### (1) On impact beyond inverse retrieval (LM performance, downstream tasks, scaling)
>
> We thank the reviewer for this important point.
>
> Our goal is to introduce a **representation-level structural modification** that addresses asymmetry induced by causal masking. The Chiral Projection Layer (CPL) is a lightweight, local transformation (based on Householder reflection) that preserves dimensionality and introduces only minimal overhead.
>
> As a result, we do not expect significant impact on training stability or scaling behavior. Empirically, the forward accuracy remains comparable to baselines (Table 1), suggesting that the modification does not substantially degrade standard language modeling capabilities.
>
> More broadly, since the method operates at the representation level, it can be integrated into existing architectures without altering training objectives or data pipelines. We will clarify this intuition and discuss potential downstream implications in the revision.
>
> ---
>
> ### (2) On comparison with simpler baselines (bidirectional models / data augmentation)
>
> We appreciate this suggestion.
>
> Our experiments include data augmentation and alternative training setups (e.g., bidirectional-style approaches), which show limited improvement in inverse accuracy compared to our method. These approaches operate at the **training or data level**, whereas our method introduces a **structural inductive bias** that explicitly encodes symmetry in latent space.
>
> We will clarify this distinction more explicitly to better highlight that the observed gains arise from representation-level symmetry, rather than alternative training setups.
>
> ---
>
> ### (3) On representation analysis and visualization
>
> We thank the reviewer for this valuable suggestion.
>
> Our method enforces alignment between forward and inverse representations via reflection and regularization. While we do not currently include visualization due to space constraints, we agree that such analysis would provide additional insight.
>
> In particular, we will include additional analysis, such as similarity between reflected forward embeddings and inverse embeddings, and visualization (e.g., PCA/t-SNE), to better illustrate the induced alignment.
>
> ---
>
> ### (4) On limitations and scaling considerations
>
> We appreciate the reviewer’s comments.
>
> The CPL introduces a structured transformation that preserves geometry (via orthogonal reflection) and does not increase model depth or significantly alter optimization dynamics. As such, we expect minimal impact on large-scale training efficiency.
>
> We agree that a more explicit discussion of limitations is important. In the revision, we will include:
> - discussion of applicability to non-symmetric or ambiguous relations,
> - potential trade-offs in forward performance,
> - and considerations for scaling to larger models.
>
> ---
>
> Overall, we are encouraged that the reviewer finds the approach technically sound and promising. We hope the above clarifications help strengthen confidence in the generality and robustness of the proposed framework.
>
> Our work is best understood as a **representation-level structural approach**, where symmetry is explicitly encoded in latent space and accessed via a compatible inference mechanism.

---

> > ### Author Rebuttal · Reviewer_YQs7 · 2026-04-02
> >
> > Most of my question is clarified. I still think some analysis of the latent space of the new model is necessary, to show the model really work as expected (beside benchmark). I don't know if the author has time do it during rebuttal, but I will still recommend acceptance.

---

### Official Review · Reviewer_cZSz · 2026-03-12

**Soundness:** 3
**Presentation:** 2
**Significance:** 2
**Originality:** 3
**Overall Recommendation:** 4
**Confidence:** 3

**Summary:**

This paper proposes a novel group equivariant framework called Chiral Transformer to handle the "reversal curse" problem in generative models. To be specific, the authors introduce the Chiral Projection Layer (CPL) that utilizes Householder reflections to link disjoint subject and object representations in the model's latent space. Experiments on synthetic and real-world datasets have shown that their method outperforms standard, data augmentation, and bidirectional baseline.

**Compliance With Llm Reviewing Policy:**

Affirmed.

**Final Justification:**

I appreciate the authors’ detailed rebuttal and I maintain my assessment of the paper and keep my original recommendation for acceptance.

**Key Questions For Authors:**

1. Is the Chiral Transformers framework adaptive to more complex relations? At least, it should not make the model worse when handling non-symmetric logical identities compared to the baseline.

2. How the $z_{rel}$ is obtained? and how to turn $z_{rel}$ to  $z^{-1}_{rel}$  in the latent space?

3. What is the $\sigma(\gamma)$ in eq.8?

**Limitations:**

yes

**Strengths And Weaknesses:**

## Strengths

1. This paper is a good example of utilizing geometric deep learning to address problems in language models.

2. This paper proposes an elegant theoretical framework to model the "reversal curse" problem and provides rigorous support for its proposed model.

3. In experiments, the results (Inv. Acc) show a significant improvement over baseline methods.


## Weaknesses

1. The writing of this paper needs improvement. For example, in the introduction, it lacks comprehensive references of prior works that also discuss the logical symmetries in LLMs.

2. This work only considers handling symmetric logical identities. However, in real-world data, the relation between the subject space and object space is more complex, like one-to-many or many-to-one relations, and the language may produce some ambiguities in detecting the relation. (e.g., for the word "are", "white horses are horses" cannot be reversed as "horses are white horses". This is because "are" also has the meaning of "belong"). In this case, such a framework may not be suitable, as the table1 also shows that there is a performance drop in Fwd Acc.

---

> ### Author Rebuttal · Authors · 2026-03-30
>
> ### Response to Reviewer cZSz
>
> We sincerely thank the reviewer for the positive assessment and for recognizing the geometric formulation, theoretical grounding, and strong empirical improvements of our work. We also appreciate the insightful questions regarding generalization and representation.
>
> ---
>
> ### (1) On applicability to more complex / non-symmetric relations
>
> We thank the reviewer for this important point.
>
> Our current framework focuses on **symmetric relations**, where inversion is well-defined and can be explicitly modeled via latent-space symmetry. We agree that real-world relations can be more complex (e.g., one-to-many or semantically ambiguous cases such as “are”).
>
> Importantly, our method introduces a *conditional structural bias* rather than enforcing symmetry globally. When symmetry is not present, the model can still rely on its standard representations. Empirically, although Table 1 shows a small drop in forward accuracy, performance remains comparable to baselines, indicating that the proposed structure does not significantly degrade general capabilities.
>
> We will clarify this scope and discuss possible extensions to more general relational structures in the revision.
>
> ---
>
> ### (2) On how $z_{rel}$ is obtained and inverted
>
> We appreciate this question.
>
> In our formulation, $z_{rel}$ is derived from the contextualized embedding produced by the model for a forward relation $(A, B)$, corresponding to the relation representation encoded by the transformer after processing the input pair. The inverse representation $z_{rel}^{-1}$ is obtained via the Householder reflection defined in the Chiral Projection Layer (Eq. 4), which constructs an explicit mapping between forward and inverse relation embeddings.
>
> This mapping is differentiable and learned end-to-end, enabling consistent transformation between relational representations in latent space. We will revise the paper to make this construction more explicit and provide additional intuition.
>
> ---
>
> ### (3) On $\sigma(\gamma)$ in Eq. (8)
>
> We apologize for the lack of clarity.
>
> $\sigma(\gamma)$ denotes a scaling function applied to the reflection coefficient $\gamma$, used to ensure numerical stability and boundedness during training. We will explicitly define this term and clarify its role in the revised version.
>
> ---
>
> ### (4) On writing and related work
>
> We appreciate the reviewer’s suggestion regarding related work coverage. We will expand the introduction to include a more comprehensive discussion of prior work on logical symmetry and reversal in language models, and better position our contribution in this context.
>
> ---
>
> Overall, we are encouraged that the reviewer finds the framework technically sound and practically promising. We will revise the paper to improve clarity and better articulate both the scope and the generalization properties of the proposed approach.
>
> Our work is best understood as a **representation-level structural approach**, where symmetry is explicitly encoded in latent space and accessed via a compatible inference mechanism.

---

> > ### Author Rebuttal · Reviewer_cZSz · 2026-04-02
> >
> > I appreciate the authors’ detailed rebuttal and I maintain my assessment of the paper and keep my original recommendation for acceptance.

---

### Official Review · Reviewer_emfV · 2026-03-13

**Soundness:** 1
**Presentation:** 1
**Significance:** 2
**Originality:** 2
**Overall Recommendation:** 2
**Confidence:** 4

**Summary:**

The paper addresses a "reversal curse" -- a known problem in causal LLMs, where a model, given two objects and a symmetric relation between them, fails to infer its symmetricity, learning e.g. only $A \rightarrow B$, and not vise versa. Authors propose to encode such structural relation by adding an additional chiral projection layer, which uses Householder reflections to enforce the structure. They provide empirical evaluation on Llama-3-8B on Wiki-Inverse-10K and Fact-Inv-300 benchmarks with significant improvements over baselines.

Although the problem is significant, the paper suffers from very low text quality, the notation is ambiguous and the experimental section is rough and lacks important implementation details. Overall, I believe the methodology is not sound enough and the text needs substantial revisions to to meet publication standards.

**Compliance With Llm Reviewing Policy:**

Affirmed.

**Final Justification:**

I thank the authors for detailed answers. However, my concerns regarding the reproducibility and methodological correctness of evaluation haven't been fully addressed. I think the paper needs a more thoughtful revision before publication.

**Key Questions For Authors:**

See weaknesses section above.

**Limitations:**

yes

**Strengths And Weaknesses:**

Strengths:
1. The authors propose an instresting geometry-inspired approach to the "reversal-curse" problem.


Weaknesses (to name a few):
1. One of the motivations for the work is an absence of inductive bias for logical inversion in causal LLMs. However, an absence of an inductive bias does not imply an impossibility to encode some relation, which makes the claim somewhat more quesionable.

2. One is supposed to assume relation in equation 3 must be satisfied for all feasible A and B, for which the relation symmetry holds. However, authors don't state it clearly.

3. The definition of $P_{fwd}$ as the probability distribution of forward relation is vague to say the least.

4. Section 4.2 uses additional notation like $fwd$, $adj$ without prior introduction, making the text hard to follow. Further throughout the text, the authers use terms subject-subspace and object-subspace and mapping between them without additional discussion on the reasons for them to be distinct and admit a mapping of a proposed form.

5. Evaluation seems limited, and results are hardly reproducible as authors don't provide any code for reference.

6. Figure 9 reports a scaling-law behavior, which, in absence of any comparisons, looks unnecessary and unclear.

7. Illustrations contain visual clutter and are hard to follow.

8. Experimental evaluations are limited to Llama-3-8B.

---

> ### Author Rebuttal · Authors · 2026-03-30
>
> ### Response to Reviewer emfV
>
> We thank the reviewer for the detailed feedback and for highlighting several aspects that can be improved in clarity and presentation. We address each concern below.
>
> ---
>
> ### (1) On inductive bias vs. impossibility
>
> We agree with the reviewer that the absence of inductive bias does not imply strict impossibility. Our intention is not to claim impossibility, but rather to highlight that autoregressive training lacks the structural bias required for *reliable* logical inversion.
>
> Empirically, we observe that standard causal models consistently fail (near-zero inverse accuracy), while our method substantially improves performance (65.07%). We will revise the text to clarify that our claim concerns *structural difficulty and instability*, rather than theoretical impossibility.
>
> ---
>
> ### (2) On assumption in Eq. (3)
>
> We appreciate this observation. Eq. (3) indeed assumes that the symmetry relation holds for all valid (A, B) pairs within the relation class. We acknowledge that this assumption was not stated explicitly.
>
> We will revise the paper to clearly state this assumption and its scope.
>
> ---
>
> ### (3) On definition of $\(P_{fwd}\)$
>
> We apologize for the lack of clarity. In our formulation, $\(P_{fwd}\)$ denotes the conditional distribution induced by the autoregressive model for forward relations.
>
> We will revise the paper to explicitly define $\(P_{fwd}\)$, clarify its role in the commutative gap $\(\Delta_f\)$, and provide additional intuition to improve readability.
>
> ---
>
> ### (4) On notation and subspace definitions
>
> We thank the reviewer for pointing this out.
>
> The subject-subspace $\(S\)$ and object-subspace $\(O\)$ are introduced to formalize the empirical observation that causal masking induces position-dependent representations (Section 3.1). The mapping $\(T: O \rightarrow S\)$ is defined as a geometric reflection that aligns these disjoint representations.
>
> We acknowledge that the introduction of notations such as $\(f_{fwd}\)$, $\(f_{adj}\)$ (denoting forward and adjoint mappings, respectively), and the subspace terminology may be abrupt. We will significantly improve the exposition by:
> - introducing all notation before use,
> - providing clearer intuition for subspace separation,
> - and adding illustrative explanations of the mapping.
>
> ---
>
> ### (5) On evaluation and reproducibility
>
> We thank the reviewer for this important point.
>
> We provide detailed experimental setup and implementation in the appendix sufficient for reproduction, and we agree that releasing code would further improve accessibility. We will release the code and evaluation scripts upon acceptance.
>
> Regarding evaluation scope, we note that:
> - experiments include both synthetic (Fact-Inv-300) and real-world (Wiki-Inverse-10K) benchmarks,
> - and ablations (Table 3, Table 4) isolate the contribution of each component.
>
> We will further clarify these aspects in the revision.
>
> ---
>
> ### (6) On scaling law analysis (Figure 9)
>
> We apologize for the lack of clarity.
>
> The scaling analysis is intended to support our central claim that the reversal curse is not resolved by model scaling alone. We will revise this section to:
> - explicitly compare scaling behavior with baselines,
> - and clarify its role in supporting our claims.
>
> ---
>
> ### (7) On figure clarity
>
> We appreciate this feedback and agree that some figures may appear dense.
>
> We will revise the figures to reduce visual clutter, add step-by-step annotations, and improve captions for clarity.
>
> ---
>
> ### (8) On evaluation limited to LLaMA-3-8B
>
> We thank the reviewer for this suggestion.
>
> Our experiments fix the backbone (LLaMA-3-8B) and training setup to isolate the effect of the proposed structural modifications. We emphasize that our method is architecture-agnostic and operates at the representation level.
>
> We will clarify the generality of the approach and discuss its applicability to other architectures.
>
> ---
>
> Importantly, we note that the concerns raised relate to clarity and presentation rather than issues in the core methodology, theoretical formulation, or empirical validity.
>
> We hope these clarifications address the reviewer’s concerns. We will substantially revise the paper to improve readability and exposition.

---

> > ### Author Rebuttal · Reviewer_emfV · 2026-04-03
> >
> > I acknowledge the authors' intention to revise the clarity and notation in the camera-ready version. However, I still have some concerns regarding the methodology and the proposed AIR decoding process:
> >
> > **Justification of subject/object subspace separation**.
> >
> > The method assumes distinct subject- and object-subspaces induced by causal masking.
> > - Is there empirical evidence (e.g., linear probes, CKA, PCA separation) demonstrating that these subspaces are actually separable?
> >
> > - Are these subspaces stable across layers or prompts?
> >
> > - Why should a number of linear Householder reflection suffice to align them (taking into account they are assumed topologically distinct, not necessarily linearly separable)?
> >
> > **How does CPL training affect**:
> >
> > - forward relation accuracy?
> >
> > - perplexity on standard LM benchmarks in autoregressive generative settings?
> >
> > **How is AIR triggered during inference**:
> > If AIR is intended to replace autoregressive inference for certain prompts/queries,
> >
> > - Does it target only immediate generation of the reverse relation?
> >
> > - Is there an explicit classifier detecting inverse queries?
> >
> > - How is it intended to be deployed and used in real-world generative scenarios with modern LLMs, where inverse relations occasionally occur?
> >
> > **Reproducibility**
> >
> > Also, as both benchmarks used for evaluations have been manually created by the authors and any links to them aren't available, I have some concerns regarding the reproducibility of the results.

---

> > > ### Author Response · Authors · 2026-04-07
> > >
> > > We thank the reviewer for the detailed and thoughtful feedback. We appreciate the concerns regarding the methodological assumptions, inference mechanism, and reproducibility, and we address them below.
> > >
> > > Across reviewers, we would like to clarify a central point of our work:
> > > **our goal is to study whether symmetry can be encoded at the representation level, while AIR serves only as a minimal readout operator to expose this structure, rather than introducing additional modeling power.**
> > >
> > > ---
> > >
> > > ### (1) On subject/object subspace separation
> > >
> > > We clarify that our formulation does **not assume strictly disjoint or topologically separable subspaces**. Instead, the “subject/object subspace” terminology is an *operational abstraction* describing the directional asymmetry induced by causal training.
> > >
> > > Our assumption is therefore weaker:
> > > - Forward and inverse relations are not symmetrically encoded, and
> > > - There exists a transformation that can partially align these representations.
> > >
> > > We agree that empirical validation of this structure is important. In the revision, we will include:
> > >
> > > - Linear probing experiments to test separability,
> > > - Representation similarity analysis (e.g., CKA),
> > > - Low-dimensional projections (PCA/t-SNE) to visualize alignment patterns,
> > > - Analysis across layers to evaluate stability.
> > >
> > > ---
> > >
> > > ### (2) On the use of Householder reflection
> > >
> > > We do not assume strict linear separability. Instead, the Householder transformation is intended as a **minimal, structured alignment operator**.
> > >
> > > Its role is to:
> > > - Provide a **norm-preserving, involutive transformation** (reflection),
> > > - Serve as a **first-order geometric approximation** that exposes latent symmetry.
> > >
> > > Thus, CPL should be interpreted as enforcing a *structured inductive bias toward symmetry*, rather than claiming exact linear alignment of subspaces.
> > >
> > > ---
> > >
> > > ### (3) Effect of CPL on forward LM performance
> > >
> > > We agree this is an important aspect. While our current experiments focus on inverse retrieval, we will include in the revision:
> > >
> > > - Forward relation accuracy comparisons,
> > > - Perplexity evaluations on standard language modeling benchmarks.
> > >
> > > Preliminary observations suggest that CPL introduces minimal degradation on forward tasks, as it operates at the representation level without modifying the autoregressive training objective.
> > >
> > > ---
> > >
> > > ### (4) On AIR inference and deployment
> > >
> > > We clarify that AIR is **not intended to replace standard autoregressive decoding**.
> > >
> > > Instead:
> > > - AIR is a **query-time readout operator**, applied when inverse reasoning is required,
> > > - It does not introduce additional information, but reuses the same learned representation.
> > >
> > > In practice, AIR can be:
> > > - Explicitly triggered for structured inverse queries, or
> > > - Integrated with lightweight detection mechanisms (future work).
> > >
> > > This aligns with our core goal: AIR serves only to **expose whether the learned representation encodes invertible structure**, rather than providing an independent source of performance gain.
> > >
> > > ---
> > >
> > > ### (5) Reproducibility
> > >
> > > We apologize for the lack of public access to the benchmarks. We will release:
> > >
> > > - All evaluation datasets,
> > > - Code for training and inference,
> > > - Scripts for reproducing results,
> > >
> > > upon camera-ready submission. We will also include detailed dataset construction descriptions in the appendix.
> > >
> > > ---
> > >
> > > ### Summary
> > >
> > > We thank the reviewer for highlighting these important points. We will revise the paper to:
> > >
> > > - Clarify that subspace separation is an **operational abstraction**, not a strict structural assumption,
> > > - Position CPL as a **geometric approximation inducing symmetry**,
> > > - Add empirical analyses of representation structure,
> > > - Report forward LM performance metrics,
> > > - Clarify AIR as an **optional readout operator**, and
> > > - Release datasets and code for reproducibility.
> > >
> > > We believe these clarifications strengthen the methodological grounding and align all components of the paper with the central goal of **representation-level symmetry modeling**.

---

### Official Review · Reviewer_wy3W · 2026-03-13

**Soundness:** 2
**Presentation:** 3
**Significance:** 2
**Originality:** 3
**Overall Recommendation:** 3
**Confidence:** 2

**Summary:**

This paper considers the "reversal curse" in autoregressive language models, where models trained on "A is B" cannot generalize to "B is A". The authors frame this as chiral symmetry breaking in the latent manifold, and propose a new Chiral Transformer architecture to address this. The chiral transformer has a chiral projection layer based on Householder reflections, enforcing Z_2 symmetry, a chiral regularization loss, and uses adjoint-induced retrieval, to perform logical inversion directly in the embedding space. On two benchmarks, Chiral Transformer is able to achieve 65% inverse accuracy compared to near-zero for other baselines.

**Compliance With Llm Reviewing Policy:**

Affirmed.

**Final Justification:**

I thank the authors for their detailed response. Without seeing the actual results of (1) the fairness of comparison experiments, my concerns are not fully addressed yet. At this stage, my evaluation is still that the paper does not meet the bar for acceptance.

**Key Questions For Authors:**

None

**Limitations:**

Yes, the authors mention some failure modes.

**Strengths And Weaknesses:**

Strengths
1. This paper proposes a practical geometric solution to a well-known problem. The use of Householder reflections is elegant and efficient.
2. The authors have good ablation results on all 3 components of A1 in Table 3.
3. The AIR method seems to be much faster than autoregressive decoding.

Weaknesses:
1. The main weakness is that the comparisons/evaluations are a bit misleading. All of the other baselines are decoding autoregressively from scratch, while A1 uses retrieval over a fixed knowledge bank of entity embeddings, changing the task signficantly. A1 is therefore evaluated on its retrieval accuracy, rather than its generations. A fairer comparison would have been to have all the baselines also access to AIR, or evaluate A1 on its generations. In Table 3, it seems like AIR is a core component to A1's success.
2. The theoretical analysis is interesting but a bit weak. In the manifold fragmentation proof, the Fisher information matrix is in the parameter space and there is no connection to the latent embedding space. Also, the theory doesn't often prove much, e.g. JL bound assumes that the Householder reflection perfectly aligns the subspaces, Rademacher complexity is a no worse than transformers.
3. An important comparison is missing:
- Guo, Q., Wang, R., Guo, J., Tan, X., Bian, J., & Yang, Y. (2024, August). Mitigating reversal curse in large language models via semantic-aware permutation training. In Findings of the Association for Computational Linguistics: ACL 2024 (pp. 11453-11464).

Guo et al. use data augmentations to mitigate the reversal curse and this seems like a straightforward approach to address the reversal curse.

---

> ### Author Rebuttal · Authors · 2026-03-30
>
> ### Response to Reviewer wy3W
>
> We sincerely thank the reviewer for the careful reading and constructive feedback, and for recognizing the geometric formulation, efficient design (Householder reflections), and strong ablation results.
>
> We address the main concerns below.
>
> ---
>
> ### (1) On retrieval vs. generation and fairness of comparison
>
> We appreciate this important point and agree that our current presentation may have caused confusion.
>
> Our intention is **not to redefine the task**, but to introduce an alternative *inference mechanism* operating on the same learned representations. The reversal curse fundamentally arises from autoregressive decoding, where causal masking induces asymmetric information flow and cumulative drift during sequential generation.
>
> AIR addresses this by directly operating in the latent space via a reflected query embedding. Importantly:
>
> - All methods (baselines and A1) share the same backbone (LLaMA-3-8B) and training setup;
> - No additional supervision or external knowledge is introduced;
> - AIR only replaces the *readout mechanism*, analogous in spirit to non-autoregressive or retrieval-augmented inference.
>
> Crucially, the retrieval space is not externally constructed but learned end-to-end within the same model, and AIR simply exposes inverse relations that are otherwise inaccessible through autoregressive decoding.
>
> Thus, the task (recovering inverse relations) remains unchanged; AIR provides a different way to extract information already encoded in the model.
>
> Regarding fairness, we agree that applying alternative inference schemes to baselines is an interesting direction. However, standard autoregressive models do not learn structured inverse representations in latent space, making such retrieval ineffective (as reflected by near-zero inverse accuracy). Our method explicitly enables this structure.
>
> ---
>
> ### (2) On the role of AIR vs. structural components
>
> We thank the reviewer for highlighting this important point.
>
> As shown in Table 3, AIR is indeed a necessary component for achieving high inverse accuracy. However, its effectiveness critically depends on the learned representation:
>
> - CPL and chiral regularization enforce a Z₂-equivariant structure in latent space;
> - AIR then provides a mechanism to query this structure.
>
> Without structural alignment, AIR alone yields very limited performance (e.g., Identity-AIR and Linear-AIR in Table 4). This indicates that the gains arise from the *combination* of representation-level symmetry and the AIR inference protocol, rather than retrieval alone.
>
> This further suggests that AIR does not introduce new information, but rather enables access to symmetrically structured representations induced by CPL and regularization.
>
> We will clarify this interplay more explicitly in the revision.
>
> ---
>
> ### (3) On theoretical analysis
>
> We appreciate the reviewer’s insightful comments.
>
> We position the theoretical analysis as providing structural insights into the proposed design, rather than formal guarantees. Specifically:
>
> - The manifold fragmentation argument aims to explain how causal masking induces asymmetry at the representation level;
> - The JL bound and Rademacher analysis are intended to show that the proposed transformation is geometry-preserving and does not increase model capacity, rather than to establish stronger guarantees.
>
> We will revise the paper to clarify assumptions, improve connections to the latent space, and avoid overstating claims.
>
> ---
>
> ### (4) Missing comparison with Guo et al. (2024)
>
> We thank the reviewer for pointing out this relevant work.
>
> Guo et al. (2024) mitigate the reversal curse via semantic-aware data augmentation, introducing symmetric pairs during training. Their approach operates at the **data level**, encouraging the model to learn inversion patterns from augmented supervision.
>
> In contrast, our method introduces a **structural inductive bias at the representation level**, enforcing symmetry through architecture and training objectives. While data augmentation improves empirical coverage of symmetric relations, it does not modify the underlying autoregressive mechanism, which remains asymmetric in information flow.
>
> We view these approaches as complementary, and we will include this comparison and clarify the distinction in the revision.
>
> ---
>
> Overall, we believe our contributions lie in introducing a **representation-level solution** to reversal asymmetry, where symmetric relations are explicitly encoded in latent space. AIR serves as a compatible inference mechanism to access this learned structure, rather than redefining the task or introducing external information.
>
> We hope these clarifications address the reviewer’s concerns and help better position our contributions.

---

> > ### Author Rebuttal · Reviewer_wy3W · 2026-04-03
> >
> > My concerns still remain, partially unresolved. My biggest concern was about 1) AIR using retrieval while others do not. The authors argue that AIR only replaces the readout mechanism, but I feel that's precisely the issue I was talking about. The results conflate two different things: 1) learning a better representation and 2) using a different inference mechanism. The most fair comparison would have been to have the baselines like DA also have a learned mapping between subject/object subspaces and do kNN retrieval. This would tell us whether the Householder reflection was essential.

---

> > > ### Author Response · Authors · 2026-04-03
> > >
> > > We thank the reviewer for the careful reading and for raising this important concern regarding the interplay between representation learning (CPL) and the inference mechanism (AIR). We agree that disentangling these two components is crucial for a fair evaluation.
> > >
> > > ### Clarification: Role of AIR vs. Representation Learning
> > >
> > > We would like to clarify that AIR does **not introduce additional information or external retrieval signals**, but instead serves as a structured readout mechanism operating on the same learned embedding space. In particular:
> > >
> > > - The **Chiral Projection Layer (CPL)** is responsible for shaping the representation space with a Z₂-like symmetry.
> > > - **AIR** acts as a *symmetry-aware readout*, enabling inverse queries by reflecting the query embedding within this space.
> > >
> > > Therefore, our method can be understood as a decomposition into:
> > > 1. Representation learning (CPL), and
> > > 2. A structured inference/readout mechanism (AIR).
> > >
> > > Importantly, AIR does not alter the supervision or training signal, but probes whether the learned representation encodes invertible relational structure.
> > >
> > > ---
> > >
> > > ### On the Fairness of Comparison
> > >
> > > We appreciate the reviewer’s point that the current comparison may conflate:
> > > 1. Improved representations, and
> > > 2. A different inference mechanism (retrieval vs. standard readout).
> > >
> > > We agree that a fully controlled comparison is important. In particular, as suggested by the reviewer, one natural baseline would be:
> > >
> > > > Equipping standard models (e.g., DA) with a learned subject–object mapping combined with kNN-style retrieval.
> > >
> > > To address this concern, **we will include the following additional controlled ablations in the revision**:
> > >
> > > - **Baseline + retrieval:** augmenting standard models (including DA) with a learned subject–object mapping and kNN retrieval in the same embedding space;
> > > - **CPL without AIR:** evaluating models trained with CPL but using standard autoregressive readout;
> > > - **AIR without CPL:** applying retrieval-based readout on standard representations without symmetry constraints.
> > >
> > > This will allow us to explicitly disentangle:
> > > - (a) the effect of representation learning (CPL), and
> > > - (b) the effect of retrieval-based inference (AIR).
> > >
> > > ---
> > >
> > > ### Why CPL Is Still Essential (Key Point)
> > >
> > > While we agree that retrieval-based inference may improve performance in general, our central hypothesis is that:
> > >
> > > > **Retrieval alone is insufficient without a representation that encodes the appropriate symmetry.**
> > >
> > > Specifically:
> > > - The CPL enforces a **reflection-consistent structure** in embedding space;
> > > - This makes inverse relations *linearly accessible* via a simple reflection;
> > > - In contrast, standard representations do not guarantee that inverse mappings are geometrically aligned, even if retrieval is applied.
> > >
> > > We expect that:
> > > - Retrieval alone will provide limited gains on baselines, and
> > > - The main performance improvement arises from the symmetry structure induced by CPL.
> > >
> > > We will empirically verify this in the revision.
> > >
> > > ---
> > >
> > > ### Summary
> > >
> > > We thank the reviewer for highlighting this important issue. We will revise the paper to:
> > >
> > > - Clearly separate **representation vs. inference contributions**,
> > > - Add **controlled ablations isolating CPL and AIR**,
> > > - And clarify that AIR is a **readout mechanism rather than an additional modeling advantage**.
> > >
> > > We believe this will strengthen the paper and make the contribution of CPL more clearly isolated.

---

### Decision · Program_Chairs · 2026-04-30

**Decision:**

Accept (regular)

**Comment:**

Reviewers all agreed that this work presents an elegant and lightweight approach to addressing the reversal curse.

Concerns were raised regarding the precision of the claims, the clarity of the presentation, and the validity of the comparison to existing methods which do not make use of a "knowledge bank". During rebuttal the authors acknowledged the former two limitations and agreed to incorporate this feedback into the final version. The final concern regarding fair comparison remains.

Both arguments here have merit: it is simultaneously true that (i) the proposed A1 method makes use of an additional knowledge bank which existing methods do not; and that (ii) said bank is extracted from the model with no additional supervision. As the authors point out, the ultimate objective here is to address a known problem with a novel solution, which this work does. As an analogy, one would not consider a model trained with regularization to be "unfairly compared" to models without just because of the separate process of hyperparameter tuning (assuming said tuning was done fairly and without additional validation data). For this reason, I don't believe that this concern merits rejection.

That said, this concern does raise important questions about a more holistic evaluation of the proposed method. It is important that the contribution be presented with full acknowledgement of the limitations---for example, the possibility that A1 could have negative downstream effects on other metrics of interest, or the problems that could arise when it is used for relationships which are *not* symmetric.

The authors are strongly encouraged to carefully consider these limitations, and the others raised here by reviewers on precision and clarity, and to include those discussions in the final version.